

# Modelling the water balance of Lake Victoria (East Africa), part 2: future projections

Inne Vanderkelen[1], Nicole P. M. van Lipzig[2], and Wim Thiery[1,3]

[1]Department of Hydrology and Hydraulic Engineering, Vrije Universiteit Brussel, Brussels, Belgium
[2]Department of Earth and Environmental Sciences, KU Leuven, Leuven, Belgium
[3]Institute for Atmospheric and Climate Science, ETH Zurich, Zurich, Switzerland

**Correspondence:** Inne Vanderkelen (inne.vanderkelen@vub.be)

**Abstract.** Lake Victoria, the second largest freshwater lake in the world, is one of the major sources of the Nile River. The outlet to the Nile is controlled by two hydropower dams of which the allowed discharge is dictated by the Agreed Curve, an equation relating outflow to lake level. Some regional climate models project a decrease of precipitation and an increase of evaporation over Lake Victoria, with potential important implications for its water balance and resulting level. Yet, nothing is known about the potential consequences of climate change for the water balance of Lake Victoria. In this second part of a two-paper series, we feed a new water balance model for Lake Victoria presented in the first part with climate simulations available through the Coordinated Regional Climate Downscaling Experiment (CORDEX) Africa framework. Our results reveal that most regional climate models are not capable of giving a realistic representation of the water balance of Lake Victoria. Therefore we applied two bias correction methods, resulting in both cases in a closed water balance. Our results reveal that for two emission scenarios (RCP 4.5 and 8.5), the decrease in precipitation over the lake and an increase in evaporation are compensated by an increase in basin precipitation leading to more inflow. The future lake level projections show that the outflow scenario and not the emission scenario is the main controlling factor of the future water level evolution. Moreover, inter-model uncertainties are larger than emission scenario uncertainties. The comparison of four different outflow scenarios for the future uncovers that the only sustainable outflow scenario is regulating outflow following the Agreed Curve. The associated outflow encompasses however large uncertainties ranging up to 177%, which are important to take into account regarding future hydropower generation and water availability downstream.

## 1 Introduction

Lake Victoria is directly sustaining 30 million people living in its basin and 200 000 fishermen operating from its shores. Therefore, the water level fluctuations of the lake are of major importance. Declining water levels may affect the local communities by their ability to access water, to fish and to transport goods (Semazzi, 2011). Further downstream, the livelihood of about 300 million people in the Nile basin is supported by the natural resources of Lake Victoria, as it is one of the two major sources of the Nile (Semazzi, 2011). Originating from Lake Victoria, the White Nile provides a more constant flow during the year, providing 70 to 90% of the total Nile discharge during the dry season in the Ethiopian highlands (Di Baldassarre et al., 2011). A drop in the water level could imply a decreasing outflow, which may have major implications downstream in



the Nile basin. The countries of the Nile basin require sufficient water resources for their future development and welbeing, considering the population growth and economic development (Deconinck, 2009; Taye et al., 2011). Consequently, there are a lot of tensions between countries in the Nile basin. The outflow of the lake is controlled by the Nalubaale and Kiira dams for hydropower. Human strategies towards regulating the outflow might therefore play a crucial role in the downstream Nile basin

water resources and associated political tensions. This will be even more relevant in the light of climate change, where water will become an even more important resource, both around the lake and downstream in the Nile Basin (Taye et al., 2011). Lake level fluctuations also influence the amount of outflow released by the dam and consequently the hydropower generated and energy available in the region. Overall, the future evolution of the levels of Lake Victoria and the outflow is vital information for the future generations living on its coasts.

The water levels of Lake Victoria are determined by the Water Balance (WB) of the lake, consisting of precipitation on the lake, lake evaporation, inflow by tributary rivers and dam outflow. Since the construction of the dam complex in 1954, a rating curve called the "Agreed Curve" was established relating the outflow and lake level in natural conditions (Sene, 2000):

$$Q_{out} = 66.3(L - 7.96)^{2.01} \qquad (1)$$

In this equation, the dam outflow $Q_{out}$ (m$^3$ day$^{-1}$) is calculated based on the lake level, $L$ (m), as directly measured at the dam.

The climate in East Africa experiences large interannual variability in precipitation (Nicholson, 2017). The region is a hotspot for climate change, as it is very likely that climate change will have a major influence on precipitation (Nicholson, 2017; Kent et al., 2015; Otieno and Anyah, 2013; Souverijns et al., 2016). In the last decades, the long rain seasons in East

Africa have experienced a series of droughts (Lyon and Dewitt, 2012; Rowell et al., 2015; Souverijns et al., 2016; Nicholson, 2016, 2017). This is in contrast with climate model projections for the upcoming decades, projecting an increase in precipitation over East Africa (Otieno and Anyah, 2013; Kent et al., 2015). This apparent contradiction has been called the East African climate paradox of which the causes remain unclear (Rowell et al., 2015). To find explanations Rowell et al. (2015) stated that more research is needed on the reliability of climate projections over East Africa, to the attribution of changing anthropogenic

aerosol emissions and to the role of natural variability in recent droughts. However, Philip et al. (2017) found that the severe drought in northern and central Ethiopia in 2015 is attributable to natural variability and therefore conclude that there is no paradox, for this type of events.

Future climate simulations with Regional Climate Models (RCMs) project a decreasing mean precipitation and an increasing

evaporation over Lake Victoria (Thiery et al., 2016; Souverijns et al., 2016). Compared to Global Climate Models (GCMs), RCMs have a high spatial resolution and are therefore able to represent regional and local scale forcings (Kim et al., 2014; Giorgi et al., 2009). In East-Africa, accounting for sub-GCM gridscale variations in topography, vegetation, lakes, soils and coastlines is of major importance, as these variations have a significant effect on the regional climate. Over the Lake Victoria



Basin (LVB), models with sufficiently high resolution are needed to reproduce key circulation features such as the lake-land breeze system (Williams et al., 2015). High resolution (∼7km grid spacing) coupled lake-land-atmosphere climate simulations for the African Great Lakes region with the Consortium for Small-Scale Modelling in climate mode (COSMO-CLM²) regional climate model are performed by Thiery et al. (2015). These simulations outperform state-of-the-art regional climate simula-

tions for Africa conducted at ∼50km grid spacing, because of the coupling to land surface and lake models and enhanced model resolution, which allows a better representation of the fine-scale circulation and precipitation patterns (Thiery et al., 2015, 2016).

At the moment, almost no research is dedicated to the potential consequences of climate change for the WB of Lake Victoria.
This is remarkable, since the evolution of the future levels of lake Victoria is vital information for the future generations living on its coasts. Tate et al. (2004) used simulations with one fully coupled GCM to model future fluctuations of the lake level. Model disparities were however very high over East-Africa and the GCM was not capable to sufficiently model precipitation over the African Great Lakes (Tate et al., 2004). Therefore, the results from Tate et al. (2004) serve as an illustration of the sensitivity of lake levels and outflows to climate change scenarios, rather than as an actual lake level projection. Recently, high
resolution ensemble climate projections became available for Africa through the Coordinated Regional CLimate Downscaling Experiment (CORDEX; Nikulin et al., 2012). Operating at 0.44° (∼50 km) horizontal resolution, these models attempt to resolve the lake and its mesoscale circulation. When these simulations are used as input for a water balance model (WBM) for Lake Victoria, future lake level projections can be generated. The ensemble approach ensures that the model spread incorporates uncertainties associated with individual model deficiencies, emission pathways and natural variability.

Here, we use the WBM constructed in the first part of this two-paper series (Vanderkelen et al., 2018a) and drive it with climate simulations from the CORDEX over the Africa domain from 1950 to 2100. First, we assess the ability of regional climate models from the CORDEX ensemble to reproduce the historical lake level of Lake Victoria. Second, a bias correction based on observations is successfully applied on these simulations. Last, the future evolution of the water level of Lake Victoria
under various climate change scenarios is investigated, together with the role of different human management strategies at the outflow dams.

## 2   Data and methodology

### 2.1   CORDEX ensemble

In recent years, RCM simulations have become available through the CORDEX framework based on the Coupled Model Intercomparison Project Phase 5 (CMIP5) GCMs according to the RCP 2.6, 4.5 and 8.5 emission scenarios (Giorgi et al., 2009). The CORDEX-Africa project provides simulations over the Africa domain, which includes the whole African continent with a spatial resolution of 0.44° by 0.44° and a daily output frequency (Nikulin et al., 2012). In CORDEX-Africa, there are currently



simulations with six RCMs available (*CCLM4-8-17, CRCM5, HIRHAM5, RACMO22T, RCA4* and *REMO2009*) for the histor-
ical period (1950-2005) and the three Representative Concentration Pathways (RCPs; 2006-2100). By using different GCMs
as initial and lateral boundary conditions, there are in total 25 historical model simulations, 11 simulations for RCP 2.6, 21
simulations for RCP 4.5 and 20 simulations for RCP 8.5 (see appendix table A1). The use of model ensembles is essential, be-

cause separate simulations show larger biases than ensemble means when they are compared to the observed climate (Nikulin
et al., 2012; Endris et al., 2013; Kim et al., 2014; Davin et al., 2016). Next to the historical and RCP simulations, the CORDEX
framework also provides an evaluation simulation for each RCM, driven by the European Centre for Medium-Range Weather
Forecasts (ECMWF) ERA-Interim reanalysis as initial and lateral boundary conditions for the period 1990-2008. Here we use
these reanalysis downscaling simulations to evaluate the skill of the RCMs by comparing them with observations.

## 2.2 Water balance model

For a detailed description of the WBM used in this paper, we refer to Vanderkelen et al. (2018a). In summary, the WB is
modelled following Eq. 2 in which the change in lake level per day $dL/dt$ (m day$^{-1}$) is calculated based on the precipitation on
the lake $P$ (m day$^{-1}$), the evaporation from the lake $E$ (m day$^{-1}$), the inflow from tributary rivers $Q_{in}$ (m$^3$ day$^{-1}$) and the dam

outflow $Q_{out}$ (m$^3$ day$^{-1}$) divided by the lake surface area $A$ ($6.83 \cdot 10^{10}$ m$^2$).

$$\frac{dL}{dt} = P - E + \frac{Q_{in} - Q_{out}}{A} \tag{2}$$

First, relevant spatial variables provided by the CORDEX simulations are regridded using a nearest neighbour remapping to
the WBM grid with a resolution of 0.065°by 0.065°(about 7 by 7 km) containing Lake Victoria and its basin. P is computed
by taking the daily mean over the lake cells of the regridded CORDEX precipitation. Basin and lake cells are defined by using

masks based on lake and basin shapefiles. E is calculated from the latent heat flux (W m$^{-2}$) simulated by the CORDEX models,
using the latent heat of vaporization, which is assumed to be constant at $2.5 \cdot 10^6$ J kg$^{-1}$. This term is aggregated in the same
way as the precipitation term. The inflow term is calculated from the daily gridded basin precipitation with the Curve Number
method as described in Vanderkelen et al. (2018a) using daily basin precipitation retrieved from the CORDEX precipitation
simulations. The same land cover classes based on the Global Land Cover 2000 data set (GLC 2000; Mayaux et al., 2003) and

the hydrological soil groups are applied to all CORDEX simulations. Note that this approach does not account for potential
influences of future land use changes on runoff. However, the Curve Number does change based on the antecedent moisture
condition for a particular day, derived from the preceding 5-day precipitation.

First, a set of WBM integrations is conducted by driving the WBM with the six CORDEX evaluation simulations. As these

evaluation simulations are driven by 'ideal' boundary conditions, this WBM simulation allows to examine the ability of the
RCMs to represent precipitation, lake evaporation, inflow and the resulting lake levels during the period when observations
for these terms are available (1990-2008). Second, the WBM is driven by the 25 historical CORDEX simulations for the



period 1951-2005. Finally, future WBM runs are conducted following the RCP 2.6, 4.5 and 8.5 scenarios. Two GCM-RCM combinations are excluded from the analysis because of inconsistencies between the historical and future simulations (see appendix A). To be comparable, the input terms to the transient WBM need to adhere to the same calendar. Therefore, the number of days are adjusted in a number of historical and future CORDEX simulations, as described in appendix B. Finally,

the WBM requires an initial lake level. The evaluation simulations start with the observed lake level in 1990 (1133.7 m a.s.l.) and the historical simulations with the observed lake level in 1950 (1134.3 m a.s.l.). For the future simulations, the last lake level calculated by the corresponding historical simulation is used.

## 2.3 Future outflow scenarios

The evaluation and historical WBM simulations use recorded outflow values. However, considering the known deviations of

water release from the Agreed Curve during the period 2000-2006 (Kull, 2006; Vanderkelen et al., 2018a), future outflow is subject to uncertainty. Therefore, four different outflow scenarios are selected based on simple assumptions. A first assumption is that future outflow starts following the Agreed Curve again. In this *Agreed Curve scenario*, daily outflow is calculated following Eq. 1 and using lake level at the dam on the previous day. Another possibility is to manage future outflow in such a way that the lake level remains constant and the water balance is closed. In this *constant lake level scenario*, daily outflow

is calculated as residual of the water balance, with the lake level kept constant at the last known lake level, ranging between 1134.5 and 1135.2 m for the different simulations. If the water balance is negative, there is no outflow, but the lake level is allowed to decrease. When the water balance is positive again, the lake level is first restored to its predefined constant height. The possible remainder from the positive water balance results then in outflow for that day. By consequence, the lake level in this scenario is constant apart from short negative excursions. In the last defined scenarios, future outflow is kept constant,

ensuring a constant supply of hydropower. Two constant outflow values are chosen based on the annual means of daily outflow values between 1954 and 2006, the period in which the dam was operational and outflow values are known. The first scenario uses the maximal annual mean outflow of $138 \cdot 10^6$ m$^3$ day$^{-1}$, measured in 1964. This scenario is therefore denoted as the *high constant outflow scenario*. The second scenario uses the minimal annual mean outflow of $50 \cdot 10^6$ m$^3$ day$^{-1}$, measured in 1955 and denoted as the *low constant outflow scenario*.

## 2.4 Bias correction method

Biases in RCM output often hamper climate impact assessments (e.g. Christensen et al., 2008; Teutschbein and Seibert, 2013; Maraun et al., 2010; Themeßl et al., 2012). Bias correction methods account for these model deficiencies by applying empirical-statistical post processing techniques such that the distribution of a certain variable from the RCM corresponds to the observed distribution. There is however no general agreement on the optimal bias correction method (Gudmundsson et al., 2012). Dif-

ferent methods exist to calculate the transformation to match the modelled variable distribution to the observed distribution for the calibration period. The main assumption of bias correction is that the bias in the RCM simulations is stationary for all scenarios (historical, RCP 2.6, 4.5 and 8.5).



In this study, we tested various bias correction methods of the *qmap* package, provided in the R language by Gudmundsson et al. (2012). The bias correction is applied on the three WB terms derived from the CORDEX simulations (daily mean lake precipitation, lake evaporation and inflow). For every historical simulation from the CORDEX ensemble, a transformation function is calculated based on the WB terms from the observational WBM presented in Vanderkelen et al. (2018a). This is done for the overlapping period of 12 years ranging from 1993 until 2005. Next, the transformation functions specific to each simulation are applied on the full historical simulation and on the available corresponding future simulations for all three RCPs. Finally, the resulting lake levels are calculated by forcing the WBM with the bias corrected WB terms.

WB closure is adhered with two of the 7 tested the bias correction methods described by Gudmundsson et al. (2012). The first method uses a linear parametric transformation to model the quantile-quantile relation between the observed and modelled data according to

$$\hat{P}_o = a + bP_m \tag{3}$$

with $\hat{P}_o$ the best estimate of $P_o$, the distribution of the observed values and $a, b$ calibration parameters. An overview of the $a$ and $b$ parameters generated for the different simulations can be found in table 1.

The second method is the non-parametric quantile mapping method, a common approach for statistical transformation (e.g. Panofsky and Brier, 1968; Wood et al., 2004; Boé et al., 2007; Themeßl et al., 2011; Themeßl et al., 2012). Following Gudmundsson et al. (2012) and Boé et al. (2007), this method uses the Cumulative Density Function (CDF) based on the empirical quantiles from the observed variable to transform the modelled variable. First, the cumulative density functions of the three WB terms following each historical simulation in the overlapping period (the reference simulations) are matched with the cumulative density function of the WB terms from the observational WBM (observations). This generates a correction function, relating the quantiles of both distributions. Next, this correction function is used to unbias the WB term simulations for the whole simulation period quantile by quantile (Boé et al., 2007).

## 3 Results

### 3.1 Evaluation water balance simulations

Results of the evaluation WBM run are compared directly to the terms used in the observational WBM (Vanderkelen et al., 2018a). Precipitation observations are retrieved from the Precipitation Estimation Remotely Sensed Information using Artificial Neural Networks - Climate Data Record (PERSIANN-CDR; Ashouri et al., 2015). Evaporation is estimated from the latent heat flux output of the high resolution reanalysis downscaling of the COSMO-CLM² regional climate model (Thiery et al., 2015).



**Table 1.** Table showing the $a$ and $b$ parameters of the linear parametric transformation for the different CORDEX simulations (Eq. 3).

| RCM | Driving GCM | Parameter $a$ | Parameter $b$ |
|---|---|---|---|
| CCLM4-8-17 | CNRM-CM5 | 27025520 | 0.257 |
| CCLM4-8-17 | EC-EARTH | 38149251 | 0.315 |
| CCLM4-8-17 | HadGEM2-ES | 47564882 | 0.404 |
| CCLM4-8-17 | MPI-ESM-LR | 36624437 | 0.261 |
| CRCM5 | MPI-ESM-LR | -1649174 | 0.705 |
| RACMO22T | EC-EARTH | -31107756 | 2.041 |
| RACMO22T | HadGEM2-ES | -16133545 | 2.225 |
| RCA4 | CanESM2 | 16021083 | 0.759 |
| RCA4 | CM5A-MR | 24386549 | 0.691 |
| RCA4 | CNRM-CM5 | 27636514 | 0.916 |
| RCA4 | EC-EARTH | 19039976 | 0.657 |
| RCA4 | GFDL-ESM2M | 36763492 | 0.652 |
| RCA4 | HadGEM2-ES | 36504161 | 1.177 |
| RCA4 | MIROC5 | 23913234 | 0.767 |
| RCA4 | MPI-ESM-LR | 21789189 | 0.812 |
| RCA4 | NorESM1-M | 31280571 | 0.946 |
| RCA4 | CSIRO-Mk3-6-0 | 22873455 | 0.971 |
| REMO2009 | HadGEM2-ES | 3349895 | 0.525 |
| REMO2009 | MPI-ESM-LR | 42228008 | 0.814 |
| REMO2009 | EC-EARTH | 40725882 | 1.013 |
| REMO2009 | CM5A-LR | 19685272 | 0.453 |
| REMO2009 | GFDL-ESM2G | 13397524 | 0.470 |
| REMO2009 | MIROC5 | 23163373 | 0.524 |

The modelled annual precipitation over the study area during the evaluation period (1990 - 2008) shows different spatial distributions (Fig. 1). Compared to the observed precipitation, the majority of the models (CCLM4-8-17, RACMO22T, RCA4 and REMO2009) underestimates the amount of lake precipitation up to -79% compared to the reference. Only HIRHAM5 gives a large overestimation of precipitation over the lake of +78% compared to the reference. The mean annual lake evaporation varies a lot among the models as well (Fig. 2). The evaporation amount is underestimated by CCLM4-8-17, RACMO22T and REMO2009 up to -71% compared to the reference and overestimated by HIRHAM5 and RCA4 up to +39 %. Similar conclusions can be drawn from the comparison of the histograms of observed and simulated lake precipitation (Fig. 3). The difference between the distributions is also found for inflow and the resulting lake levels. The seasonal cycles of the water balance terms, lake precipitation, evaporation and inflow (Fig. 4b, c and d) show the same over- and underestimations com-



pared to the observed values. This indicates that even RCMs downscaling reanalysis data still entail important precipitation and evaporation biases. In most cases, the biases in precipitation and evaporation result in drifting lake levels (figure 4a). The overestimation of HIRHAM5 in the precipitation term is too large to be compensated by its overestimated evaporation term. The modelled HIRHAM5 lake levels shows therefore an unrealistic increase. The lake levels modelled with CCLM4-8-17,

RACMO22T, RCA4 and REMO2009 show a large, unrealistic drop up to 13.3 m, which is mainly due to the underestimation of lake precipitation and inflow. Only CRCM5 is able to represent the lake level within an acceptable range.

Despite the fact that the WB based on most RCMs is not closed, we quantified the future change of the three WB terms according to the three RCP scenarios for all simulations. This is achieved by computing the difference between the future

(mean of the period 2071-2100) and the historical (mean of the period 1971-2000) simulations (Fig. 6). The climate change signal for lake precipitation differs between the simulations in every RCP scenario (Fig 6a, b, c). For some simulations, lake precipitation demonstrates a strong decrease (e.g. CCLM4-8-17 driven by CNRM-CM5, CCLM4-8-17 driven by EC-EARTH, and REMO2009 driven by EC-EARTH) while other simulations show a strong increase (e.g. REMO2009 driven by CM5A-LR). The model simulations with a moderate increase or decrease also vary in sign. In contrast to lake precipitation, the lake

evaporation signal is more consistent for the different simulations, with generally an increasing trend (Fig. 6d, e and f). The future changes in the inflow term are generally consistent as well and show an increase of inflow under all three RCP scenarios. As lake inflow is directly related to precipitation, the increase in inflow can be attributed to the increase of precipitation over the LVB.

Some of the model simulations provide extreme and/or contrasting change signals for every WB term and RCP for which they are available (CCLM4-8-17 and REMO2009 driven by EC-EARTH and REMO2009 driven by MPI-ESM-LR). This could be attributed to a inadequate model performance over the study area. The mixed climate change signal of lake precipitation can be the reflection of differences in quality of the representation of lake surface temperature and by consequence of the local dynamics induced by the diurnal temperature difference between lake and land. The bias in lake surface temperature appears

to be not that important for correct estimations of the evaporation signal, but impedes resolving the local convection and associated precipitation.

## 3.2  Simulations with bias corrected water balance terms

To be able to use all CORDEX simulations, two bias correction methods are applied. After applying the linear parametric

transformation on the CORDEX evaluation simulations, the seasonal cycle of the lake precipitation, lake evaporation and inflow term approximates the observations (Fig. 5). Consequently, the resulting lake levels lie within the range of observed lake levels. As the linear parametric bias correction method provides a closed WB for all six RCMs, it is applied on the WB terms of the historical and future CORDEX simulations. Using the second bias correction method, with empirical quantiles, also leads to WB closure. Applying this bias correction method on the historical and future CORDEX simulations yields similar results



as the linear parametric transformation. Therefore, only results from the linear parametric transformation are shown hereafter. Results with the empirical quantile bias correction method are presented in the appendix.

After applying the linear parametric bias correction, the sign of climate change signals is generally preserved, except for the
evaporation term. The amplitude of the changes generally decreases (Fig 7). This decrease is larger for the simulations with the more extreme signals, with important effects on the multi-model means.

### 3.3    Future lake level and outflow projections

Lake level projections following different outflow scenarios are computed from the CORDEX simulations, bias corrected with
a linear parametric transformation (Fig. 8). Future lake levels according the constant lake level scenario are not represented in this figure, as they are constant at their 2006 level per definition. Under the high constant outflow scenario, the ensemble mean projects a lake level decline of 27.9 m for RCP 2.6, 29.2 m for RCP 4.5 and 27.3 m for RCP 8.5 in 2100, compared to the 2006 level (Fig. 8a). The decrease is due to the outflow being too high to keep the lake levels fluctuating within the historical range. The imbalance accumulates over the years, leading to the large drop in lake level. The model uncertainty increases over
time for the three RCP scenarios, with a largest increase for RCP 8.5 and the eveloped width ranging up to 47.8 m by 2100. Hence, the different simulations tend to diverge and the projections encompass large uncertainties. Similar results are found for the low constant outflow scenario, but in this scenario, there is an increase in lake level up to 16.8 m for RCP 2.6, 15.5 m for RCP 4.5 and 17.4 m for RCP 8.5 (Fig. 8b). The model uncertainty increase for the three RCPs is similar compared to the high constant outflow scenario.

In the Agreed Curve scenario, the outflow is adjusted every day based on the lake level of the previous day. In this scenario, the modelled lake levels stay within the range of the historical fluctuations and show no clear trend (Fig. 8c). The seasonal cycles in lake level are clearly visible in the ensemble mean. In 2100, the uncertainty has increased to 1.2 m for RCP 2.6, 2.7 m for RCP 4.5 and 4.2 m for RCP 8.5. It is not surprising that the lake level modelled with this scenario stays within the historical
range, as the approximated WB equilibrium is maintained by adjusting the outflow based on the lake level on a daily basis. Moreover, the Agreed Curve relation between outflow and lake level is originally made to mimic natural outflow, accounting for the natural climate variability (Sene, 2000).

Outflow projections for the 21st century are computed for the constant lake level scenario and the Agreed Curve scenario
(Fig. 9). In the constant outflow scenario, the amount of outflow varies following a spiky pattern. Since the outflow is altered each day to maintain a constant lake level given the precipitation, inflow and evaporation terms of that day, the outflow amount greatly varies on daily time scales with an average standard deviation of $129 \cdot 10^6$ m$^3$ day$^{-1}$. Figure 9a therefore shows annually averaged daily outflow values. The three RCPs show no clear trend, but uncertainties range up to $204 \cdot 10^6$ m$^3$ day$^{-1}$. Fluctuating around a multi-model mean of $89 \cdot 10^6$ m$^3$ for RCP 2.6, $88 \cdot 10^6$ m$^3$ for RCP 4.5 and $94 \cdot 10^6$ m$^3$ for RCP 8.5, annual average





outflow amounts lie in the same range as historical outflow amounts, with an average outflow amount of $88 \cdot 10^6$ m$^3$ day$^{-1}$, measured from 1950 until 2006.

Outflow following the Agreed Curve scenario has no clear trend, similar to its lake levels (Fig. 9b). Compared to the outflow

for the constant lake level scenario, the inter-annual outflow variation following the Agreed Curve is smaller, as it has an average standard deviation of $7 \cdot 10^6$ m$^3$ day$^{-1}$. The maintained outflow, which does not vary a lot, is similar to the observed historical outflow values. However, the range of the projections is very large, increasing up to $161.5 \cdot 10^6$ m$^3$ day$^{-1}$ for RCP 8.5, almost double compared to the projected multi-model mean ($91,3 \cdot 10^6$ m$^3$ day$^{-1}$). This has important implications for the potential hydropower generation and water availability downstream, as the lowest percentiles are close to zero, representing

limited outflow amounts.

## 4  Discussion

None of the RCMs, except for CRCM5, are able to provide reliable estimations of the water balance terms in the LVB (Fig 3 and Fig. 4). Endris et al. (2013) found that most RCMs simulate the main precipitation features reasonably well in East-Africa. Over the whole CORDEX-Africa domain, all RCMs capture the main elements of the seasonal mean precipitation distribution and its cycle (Nikulin et al., 2012; Kim et al., 2014). Yet all these model evaluation studies also acknowledge that significant

biases are present in individual models depending on season and region. Here, a specific region is investigated wherein lakes act as main driving features of the regional climate (Thiery et al., 2015, 2016, 2017; Docquier et al., 2016). The performance of the models over Lake Victoria and its basin is primarily determined by how lakes are resolved in the models. Therefore, a correct representation of lake surface temperatures is crucial to account for the lake-climate interactions and associated mesoscale

circulations (Stepanenko et al., 2013; Thiery et al., 2014a, b). Both RCA4 and CRCM5 represent lakes through the one dimensional lake model FLake (Samuelsson et al., 2013; Hernández-Díaz et al., 2012; Martynov et al., 2012). The other CORDEX models have no lake model embedded. In RACMO22T for example, lake temperatures are parametrized by a simple algorithm which links lake surface temperatures with deeper soil temperature fields of land points at the lake shore (van Meijgaard et al., 2008). In CCLM4-8-17 on the other hand, lake surface temperatures are interpolated sea surface temperatures, typically char-

acterized by a cold bias leading to suppression of night-time convection and precipitation. The fact that RCA4, which has a lake model, gives no accurate representation of the water balance terms in the Lake Victoria Basin, is most likely due to other model biases apart from the lake model.

The precipitation signal towards the end of the century following RCP 8.5 is spatially consistent with the precipitation singal

reported by Souverijns et al. (2016) based on CORDEX simulations as well, with on average a decrease of precipitation over the Lake Victoria and an increase of precipitation over the LVB. However, the amount of decrease in lake precipitation in this study does not correspond. Here, the future change in the non-bias corrected lake precipitation term for RCP 8.5 (0.14 mm day$^{-1}$) is small compared to the values found by Souverijns et al. (2016) (up to 0.5 mm$^{-1}$) over the lake). This is attributable to




differences in CORDEX ensemble members. Compared to the ensemble of Souverijns et al. (2016), we excluded the simulation with HIRHAM (see Sect. A) and included three new simulations with REMO2009 (driven by CM5A-LR, EC-EARTH and MPI-ESM-LR) and the RCA4 CSIRO-Mk3-6-0 simulation. The new ensemble composition appear to weaken the precipitation signal.

Thiery et al. (2016) performed high resolution simulations (∼7km grid spacing) with the coupled land-lake-atmosphere model COSMO-CLM² under RCP 8.5. In these simulations, the precipitation shows a decrease of -7.5% towards the end of the century over the lake surfaces of the African Great Lake region, which is a higher decrease than found in this study (-1.5%). Precipitation over Lake Victoria is mainly generated by the lake-land breeze induced mesoscale circulation. As future

projections with COSMO-CLM² reveal a higher increase of night-time air temperature over land compared to the lake, the night-time temperature difference between land and lake decreases. This results in a weakened land breeze, responsible for the moisture advection and updrafts above the lake. Furthermore, the lake breeze and associated moisture divergence from the lake during the day is intensified by the higher increase in temperature over land compared to the lake (Thiery et al., 2015, 2016; Souverijns et al., 2016). The increase in lake evaporation following RCP 8.5 according to Thiery et al. (2016) confirms the sign

of the evaporation signal of this study, but the non-bias corrected multi-model mean (+49 mm year$^{-1}$) is lower compared to the value reported by Thiery et al. (2016) (+ 142 mm year$^{-1}$). This difference likely emerges from the negative signal of three CORDEX simulations, affecting the multi-model mean (Fig. 6f). Physically, the increase in evaporation is related to the projected increase in temperature. As evaporation is determined by both temperature and moisture availability, the higher increase in evaporation over the lake compared to surrounding land can be explained by the unlimited availability of water over the lake

surface. The differences between COSMO-CLM² and the CORDEX ensemble are an example of the fact that simulations with COSMO-CLM² outperform CORDEX-Africa simulations in general, because of its enhanced resolution (0.0625° compared to 0.44°). COSMO-CLM² simulations are however only available in time slices, and could therefore not be used to feed the WBM.

From the CORDEX ensemble, the CRCM5 RCM performed best in the evaluation simulations, as it is the only model with

a closed water balance (Fig. 4a). The only considered CRCM5 projection is driven by MPI-ESM-LR under RCP 4.5. The signals for evaporation and basin precipitation of this simulation correspond to the COSMO-CLM² simulation (Thiery et al., 2016), whereas the decrease in lake precipitation in the COSMO-CLM² simulation is not present in the CRCM5 simulation (Fig 6b). Yet, the good performance of CRCM5 can be attributed to the presence of the lake model Flake, ensuring a realistic representation of lake surface temperatures.

The decrease in lake precipitation for RCP 4.5 and 8.5 is not visible in the lake level projections following the Agreed Curve or the constant lake level scenario. This is due to the fact that the decrease in lake precipitation is compensated in the total WB by the increase in lake inflow, which is determined by the increase in precipitation in the LVB. In the bias corrected simulations using the linear parametric transformation, the deficit in lake precipitation is compensated by an increase in inflow for 68 %

©c Author(s) 2018. CC BY 4.0 License.





(RCP 4.5) and 131 % (RCP 8.5). In the RCP 2.6 scenario, this effect is not present.

Applying a bias correction on the WB terms of the CORDEX simulations was necessary to be able to make lake level and outflow projections. RCMs are often bias corrected, as their simulations inhibit errors (Christensen et al., 2008; Teutschbein
and Seibert, 2013; Maraun et al., 2010; Themeßl et al., 2012). In this study, both linear parametric transformation and the quantile mapping bias correction methods are used. The advantage of the latter is the simplicity and transparency of the method (Teutschbein and Seibert, 2013). The quantile mapping method on the other hand, is a non-parametric method and is able to correct for errors in variability as well (Themeßl et al., 2011). Yet, no substantial differences could be noted between the resulting projections of both methods, which supports that there is no single optimal way to correct for RCM biases (Themeßl et al.,
2011). In this study, the bias correction serves its purpose. It is however important to consider the limitations concerning the bias correction methods. In both methods, each WB term is corrected independently, whereas biases may not be independent among the terms, which may be important in the context of climate change (Boé et al., 2007). The consistency between the variables could be preserved by using a more sophisticated method using a multivariate bias correction (Cannon, 2017; Vrac and Friederichs, 2015). However, Maraun et al. (2017) showed that bias correction could lead to improbable climate change
signals and cannot overcome large model errors.

The analysis reveals that the outflow scenario has an important influence on the future lake levels. In the high and low outflow scenarios, the lake level drastically decreases and increases, reflecting the imbalance in the WB caused by the outflow. This imbalance is not present in the Agreed Curve scenario, where the lake level and outflow adjust based on the climatic factors
and stay both within the historical observed range. Providing a constant hydropower supply, which implies a constant outflow, may lead to to unnatural variations in lake level by letting it increase or decrease to previously unknown levels. Accompanied with these large changes in lake level, the lake extent will alter substantially, causing various impacts along the shoreline like affecting the accessibility of fishing grounds and harbours located in shallow bays. Furthermore, to meet a constant lake level, the outflow values have to be highly variable, which is not realistic if a more or less constant hydropower generation is pursued.

If the released dam outflow follows the Agreed Curve, the lake level will reflect the climatic conditions and it will fluctuate within its natural range. Moreover, the corresponding outflow following the multi-model mean stays also within the observed range. However, the range of the outflow simulations reaches up to 177% of the projected multi model mean, highlighting that future lake level trajectories may strongly differ even under a single management scenario. Nevertheless, considering the
multi-model mean, the Agreed Curve scenario can be denoted as a sustainable outflow scenario. However, violations against the prescribed outflow can have important consequences for the lake levels, as shown by the observed drop in lake level in 2004-2005, which was for 48% attributable to an enhanced dam outflow (Vanderkelen et al., 2018a). In Uganda, hydropower provides up to 90% of the energy generated (Adeyemi and Asere, 2014). There is a rapidly growing gap between energy supply and a rising demand, as the energy consumption per capita in Uganda is among the lowest in the world. The Kiira
and Nalubaale hydropower stations, managing Lake Victoria's outflow, are the largest energy sources in the country (Adeyemi





and Asere, 2014). The third largest capacity is provided by the new Bujagali hydropower dam located 8 km downstream of Lake Victoria. Therefore, operations at those dams will become even more important in the future. If there are again violations against the Agreed Curve because of the increasing hydropower demand, this may have substantial consequences for the future evolution of the lake level. A relative stable lake level is however necessary for local water availability providing resources to

the 30 million people living in its basin and to the 200 000 fisherman operating from its shores (Semazzi, 2011).

In all outflow scenarios, the model uncertainty appears to be larger than the uncertainty related to the emission scenario. This could be seen by the large spread around the multi-model mean and the coinciding RCP curves and spread (Fig. 8 and 9). The spread according to the RCP 2.6 scenario is the smallest. This scenario contains only 11 simulations, while there are 19

simulations following RCP 4.5 and 17 simulations following RCP 8.5. The future projections provide no clear differentiation between the three RCP scenarios, indicating that uncertainties associated with the model deficiencies play a more important role. Therefore, to further refine lake level projections presented in this study, it is of vital importance to account for model deficiencies.

Apart from the large model uncertainties, this approach using the WB model has some other shortcomings. First, we do not account for future land cover changes in the inflow calculations, as a static land cover map for the year 2000 is used (Vanderkelen et al., 2018a). Changes in land cover are however very important in future simulations, as they affect the CN number and therefore the amount of runoff on the one hand (Ryken et al., 2014). Moreover, future changes in land use could induce changes in precipitation in tropical areas (Akkermans et al., 2014; Lejeune et al., 2015). Second, the employed outflow

scenarios are based on three simple assumptions. The outflow scenario exerts a major influence on future lake level fluctuations and future lake levels appeared to be sustainable only if the Agreed Curve is followed. As the importance of dam management in response to rising hydropower demand increases, more sophisticated outflow scenarios accounting for the rising hydropower demand could be developed and examined whether they provide sustainable lake levels.

## 5   Conclusions

In this second part of a two paper series, a water balance model constructed developed for Lake Victoria is forced with climate projections from the Coordinated Regional Climate Downscaling Experiment (CORDEX) ensemble following three Representative Concentration Pathways (RCPs 2.6, 4.5 and 8.5) to derive projections of the future lake level fluctuations of Lake Victoria. Our results show that regional climate models incorporated in the CORDEX ensemble are typically not able to reproduce Lake Victoria's water balance. As their balance is not closed, these regional climate projections cannot be directly used to

project future lake levels. The lack of a lake model embedded in the regional climate model on the one hand, and the relative coarse resolution of the climate simulations on the other hand, cause the mesoscale circulation in the Lake Victoria basin to be not well resolved.



Accordingly, the three water balance terms provided by the CORDEX simulations (lake precipitation, lake evaporation and inflow) are bias corrected using two different bias correction methods: a parametric linear transformation and empirical quantiles. The choice of bias correction method appeared to have little influence on the results. It appears that the decrease in lake precipitation in RCP 4.5 and 8.5 is compensated by an increase in inflow, which is directly determined by precipitation in the
Lake Victoria basin.

Lake level fluctuations are projected up to 2100 using four different outflow scenarios, accounting for different dam operating strategies. Our results show that the choice of the management strategy will determine whether the lake level evolution remains sustainable or not. The two constant outflow scenarios, providing a constant hydropower supply, lead to drifting lake
levels. Furthermore, in the constant lake level scenario, outflow amounts are highly variable, which is not realistic regarding the production of hydropower. When the Agreed Curve is followed, the scenario in which outflow is adjusted based on the lake level to mimic natural outflow, the evolution of lake level and outflow remain sustainable for most realizations. However, there are large uncertainties related to the outflow projections, ranging up to 177%. Fluctuating within the historical range, the multi-model mean projected lake levels following the Agreed Curve show no clear trend. Next to the outflow scenario, we
found that the uncertainty related to the model simulations is larger than the uncertainty related to the emission scenario.

In this study, we provide the first indications of potential consequences of climate change for the water level of Lake Victoria. The large biases and uncertainty present in the projections stresses the need for an adequate representation of lakes in RCMs to be able to make reliable climate impact studies in the African Great Lakes region. Finally, the evolution of future lake levels of
Lake Victoria are primarily determined by the decisions made at the dam. Therefore, the dam management of Lake Victoria is of major concern to ensure the future of the people living in the basin, the future hydropower generation and water availability downstream.



**Figure 1.** Annual accumulated precipitation during the period 1993-2008 as derived from PERSIANN-CDR (a) the CORDEX evaluation simulations (b-g).





**Figure 2.** Annual accumulated evaporation during the period 1993-2008 as derived from COSMO-CLM$^2$ (a) the CORDEX evaluation simulations (b-g).





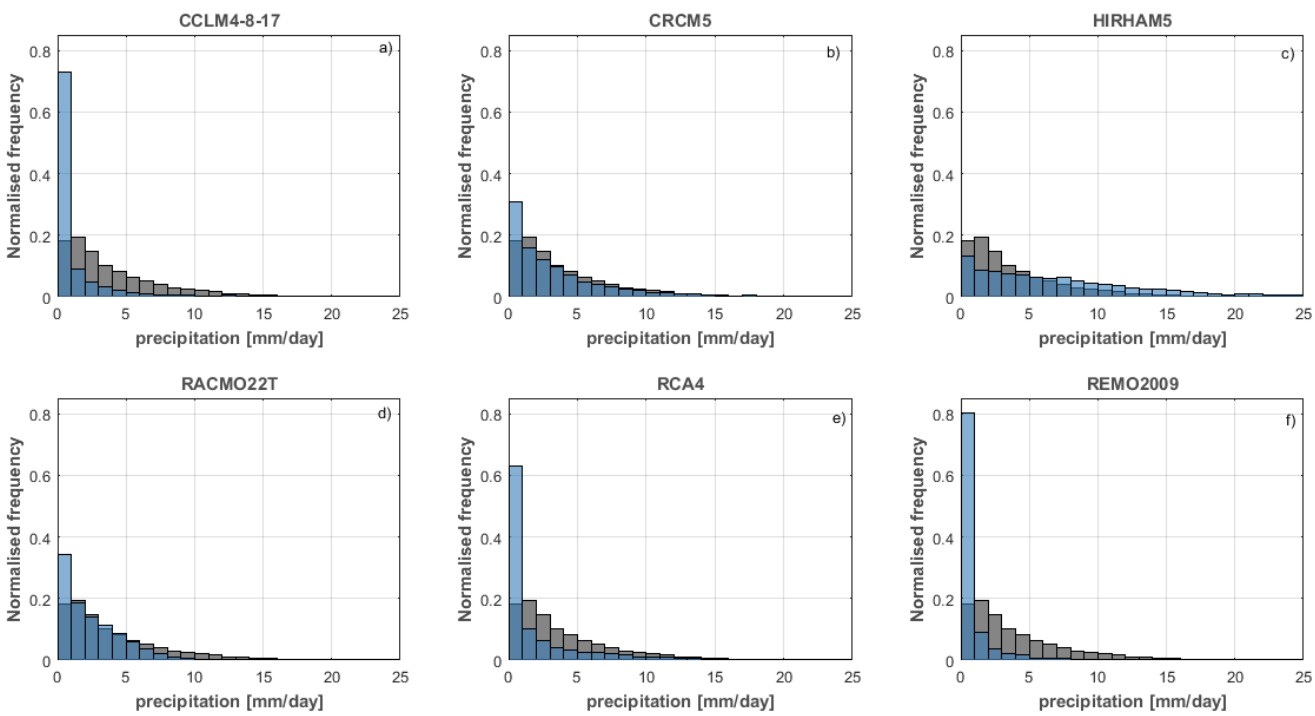

**Figure 3.** Histograms of lake precipitation derived from the CORDEX evaluation simulations for the period 1993-2008 (observed distributions, derived from PERSIANN-CDR are indicated in grey).





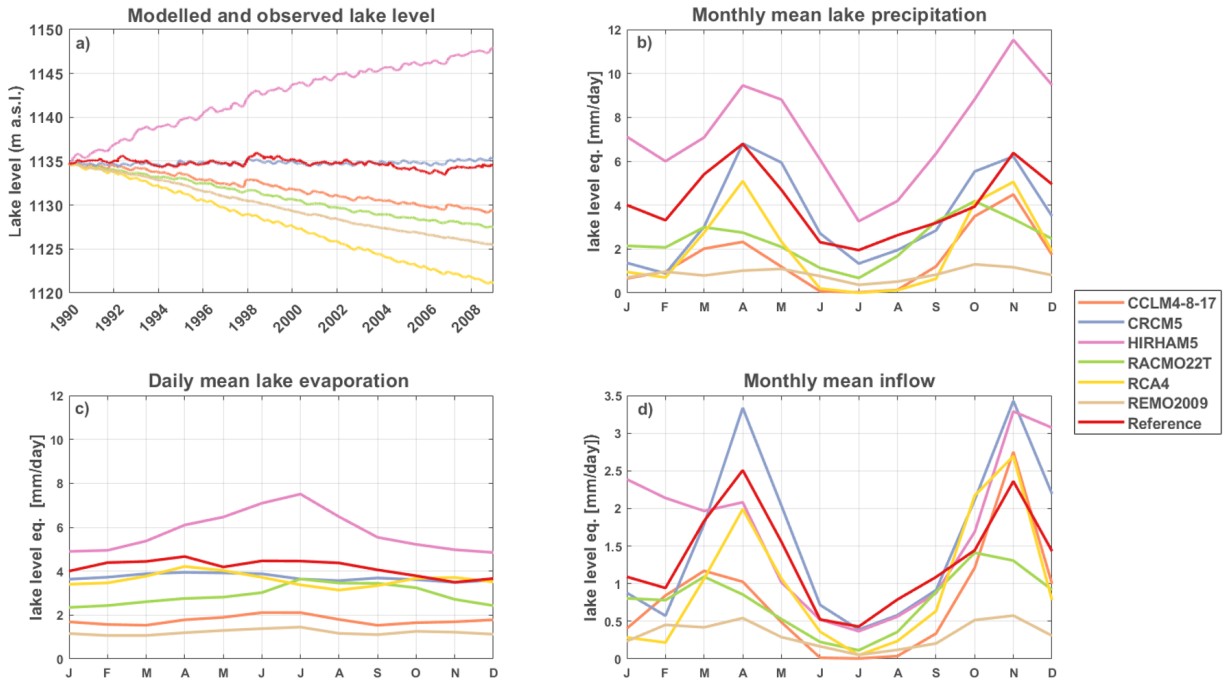

**Figure 4.** Modelled lake levels (a), seasonal precipitation (b), evaporation (c) and inflow (d) according to the CORDEX evaluation simulations without bias correction. Note the different y-axis scales.





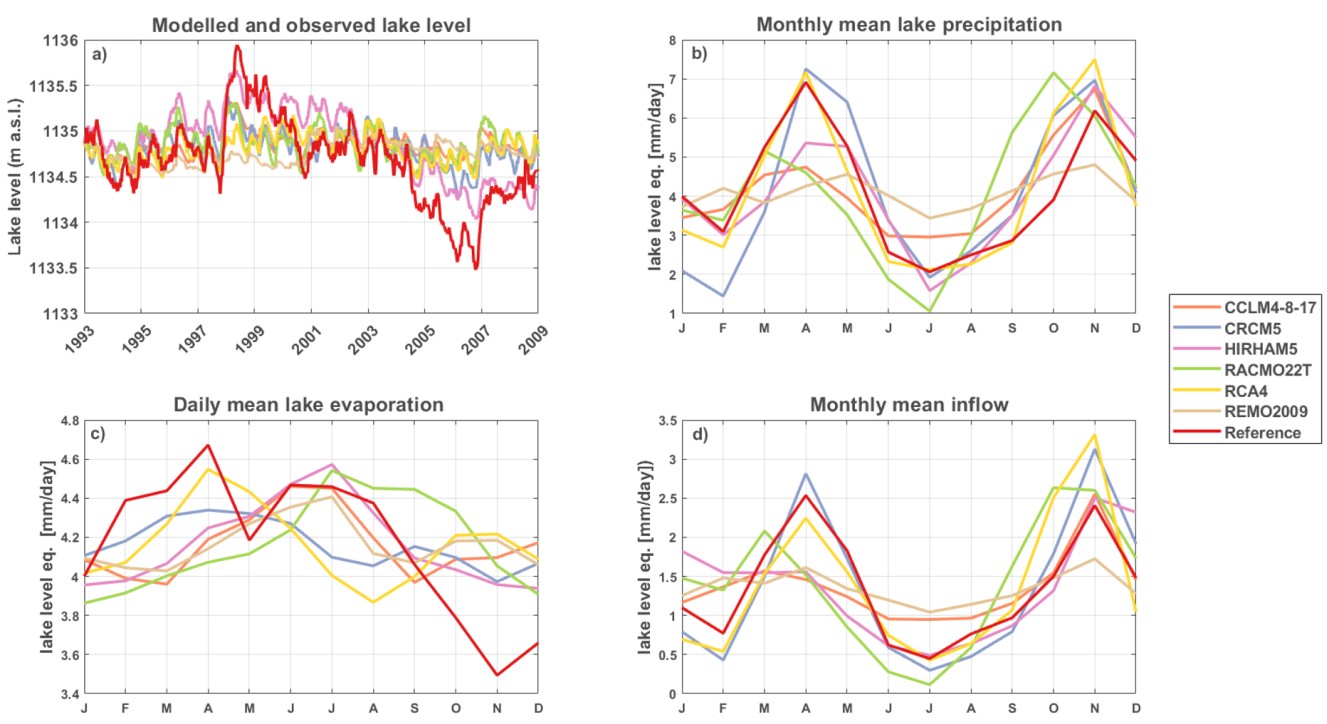

**Figure 5.** Modelled lake levels (a), seasonal precipitation (b), evaporation (c) and inflow (d) according to the CORDEX evaluation simulations, bias corrected using parametric linear transformations. Note the different y-axis scales.





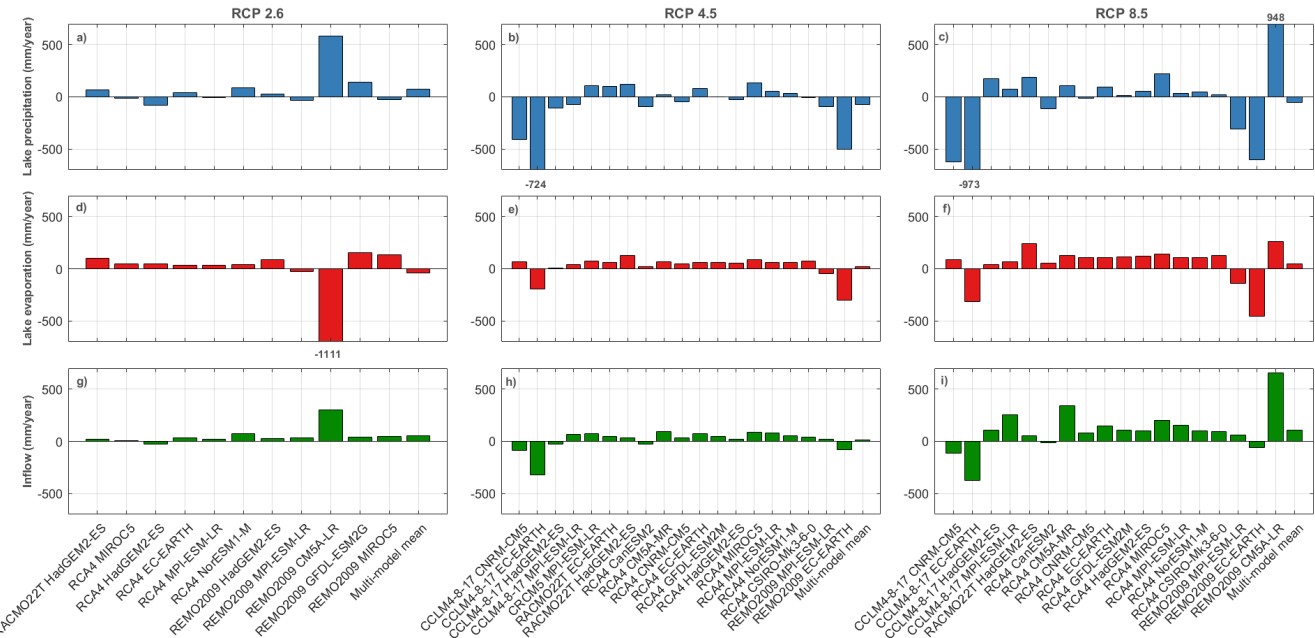

**Figure 6.** Barplots showing the projected climate chance following RCP 2.6, 4.5 and 8.5 for lake precipitation (a-c), lake evaporation (d-f) and inflow (g-i) for the CORDEX simulations without bias correction. The climate change signal is defined as the difference between the historical (1971-2000) and the future (2071-2100) simulations.





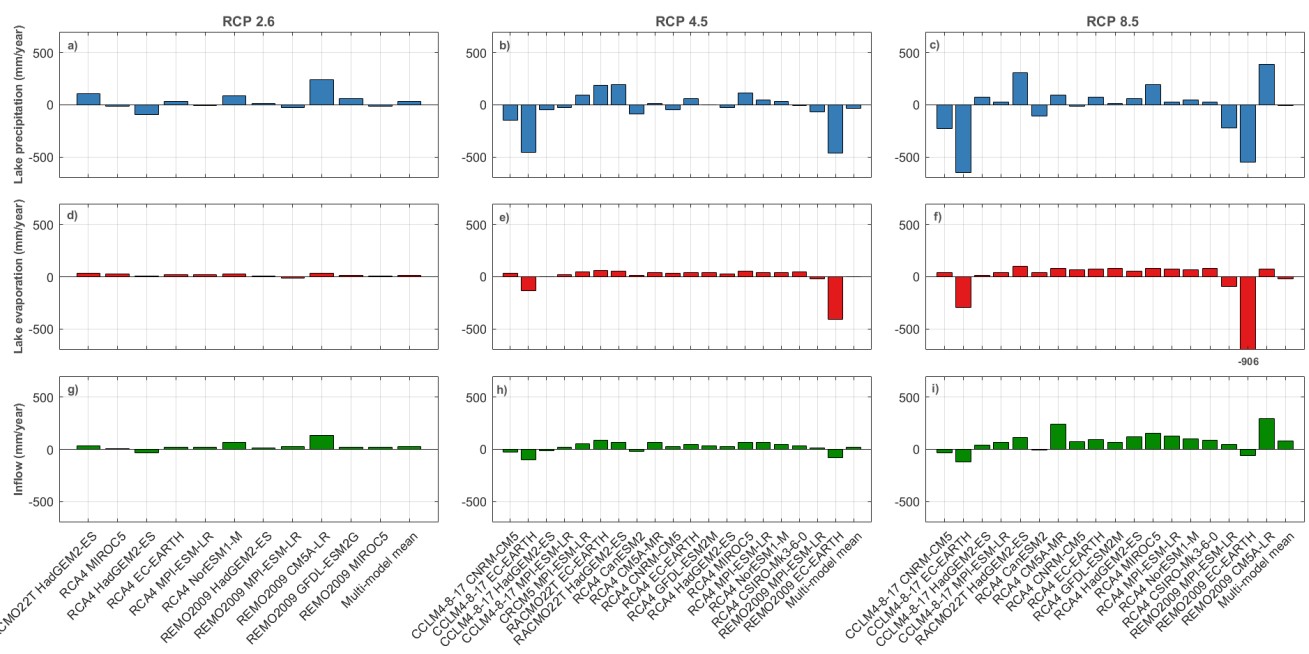

**Figure 7.** Barplots showing the projected climate chance following RCP 2.6, 4.5 and 8.5 for lake precipitation (a-c), lake evaporation (d-f) and inflow (g-i) for the CORDEX simulations bias corrected with the linear parametric transformation. The climate change signal is defined as the difference between the historical (1971-2000) and the future (2071-2100) simulations.





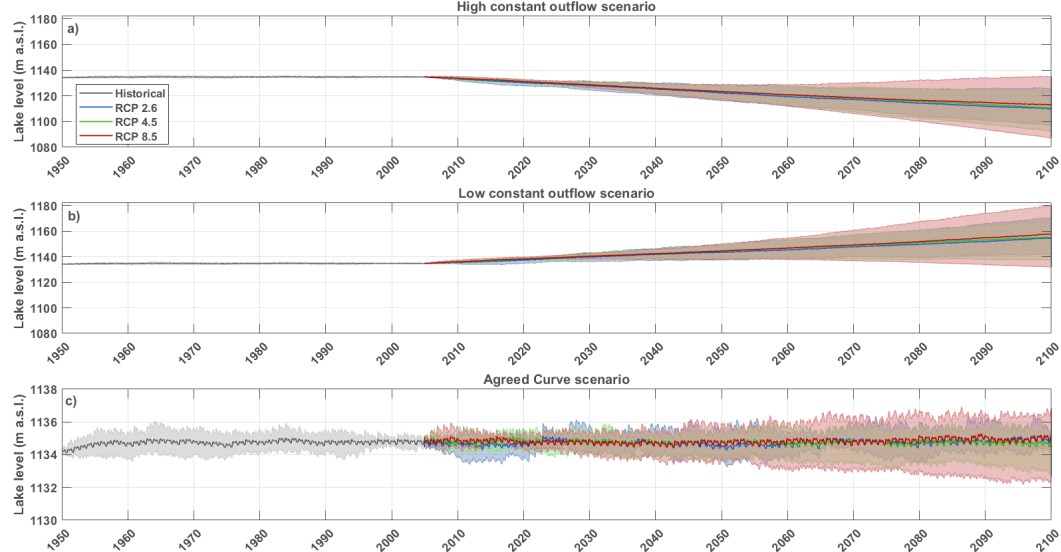

**Figure 8.** Lake level projections for the *high constant outflow scenario* ($138 \cdot 10^6$ m$^3$ day$^{-1}$) (a), the *low constant outflow scenario* ($50 \cdot 10^6$ m$^3$ day$^{-1}$) (b) and the *Agreed Curve scenario* (c). The full line shows the ensemble means and the envelope the 5th - 95th percentile of the CORDEX ensemble simulations, bias corrected using the parametric linear transformation method.

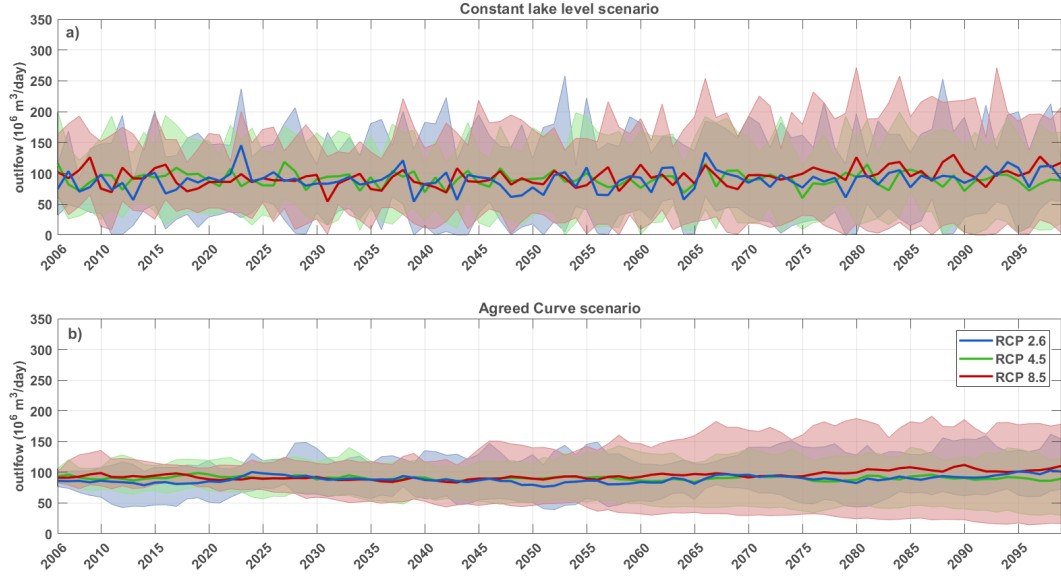

**Figure 9.** Outflow projections (annual averaged) for the *constant lake level scenario* (a) and the *Agreed Curve scenario* (b). The full line shows the ensemble means and the envelope the 5th - 95th percentile of the CORDEX ensemble simulations, bias corrected using the parametric linear transformation method.



*Code and data availability.* The Water Balance Model code is publicly available at https://github.com/VUB-HYDR/WBM_LakeVictoria. The qmap R-package is available on the Comprehensive R Archive Network (https://cran.r-project.org/). Data from the Coordinated Regional Climate Downscaling Experiment (CORDEX) Africa framework is available at http://cordex.org/data-access/esgf/

## Appendix A: Details on used CORDEX simulations

5   The CORDEX ensemble simulations used in this study are listed in table A1. From all available simulations, HIRHAM5 driven by EC-EARTH and CRCM5 driven by CanESM2 are not used due to the fact that they exhibit discrepancies between their historical and future simulations. For HIRHAM5 EC-EARTH, the absolute values of the future projections for both RCP scenarios of the three original WB terms are higher than the (more realistic) historical simulation (Fig. A1). For CRCM5 CanESM2 the historical simulations of the WB terms cause the gap, containing higher values than the observed range. These
10  discrepancies are unphysical and inhibit the application of a bias correction. Therefore, these simulations are excluded in the analysis.

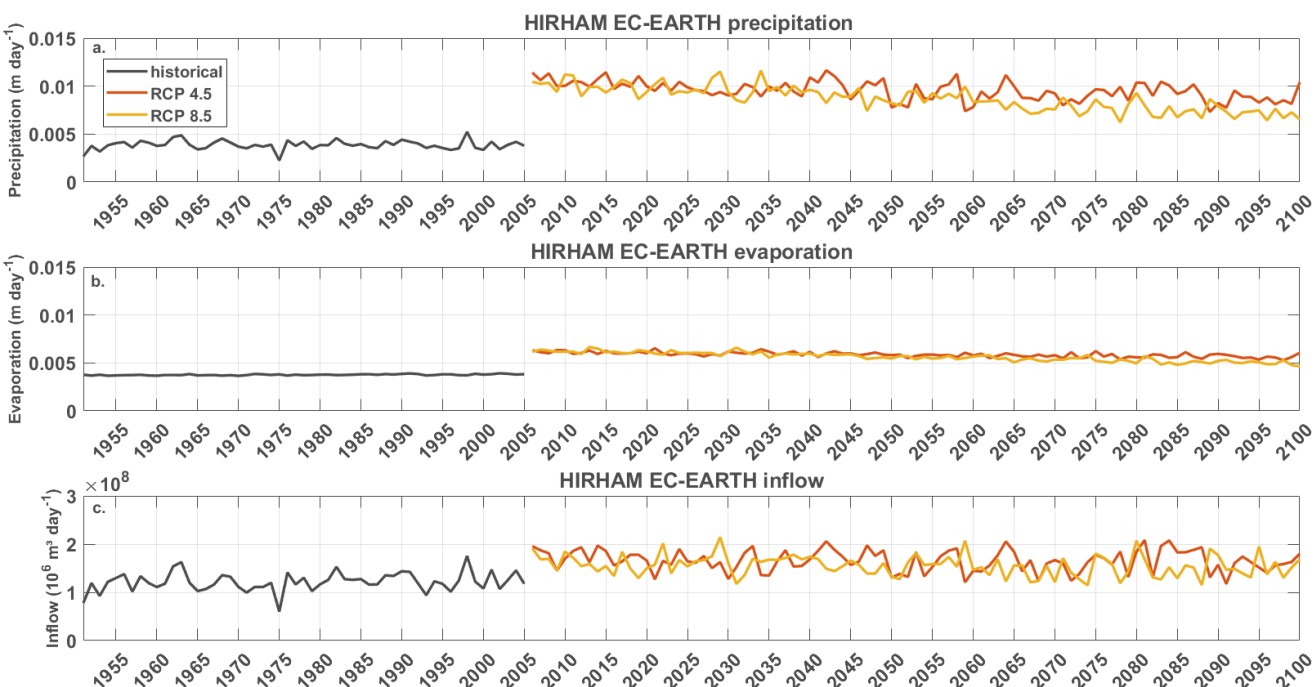

**Figure A1.** WB terms following historical, RCP 4.5 and 8.5 of HIRHAM5-EC-EARTH simulations (without bias correction).



**Table A1.** Overview of the different CORDEX simulations and their availability. (* data not used because of discrepancies between historical and future simulations).

| RCM | Driving GCM | RCP 2.6 | RCP 4.5 | RCP 8.5 |
|-----|-------------|---------|---------|---------|
| CCLM4-8-17 | EC-EARTH | N | Y | Y |
| CCLM4-8-17 | HasGEM2-ES | N | Y | Y |
| CCLM4-8-17 | MPI-ESM-LR | N | Y | Y |
| CCLM4-8-17 | CNRM-CM5 | N | Y | Y |
| CRCM5 | MPI-ESM-LR | N | Y | N |
| CRCM5 | CanESM2 | N | Y* | N |
| HIRHAM5 | EC-EARTH | N | Y* | Y* |
| RACMO22T | EC-EARTH | N | Y | Y |
| RACMO22T | HadGEM2-ES | Y | Y | Y |
| RCA4 | CanESM2 | N | Y | Y |
| RCA4 | EC-EARTH | Y | Y | Y |
| RCA4 | MIROC5 | Y | Y | Y |
| RCA4 | HadGEM2-ES | Y | Y | Y |
| RCA4 | NorESM1-M | Y | Y | Y |
| RCA4 | GFDL-ESM2M | N | Y | Y |
| RCA4 | CM5A-MR | N | Y | Y |
| RCA4 | CNRM-CM5 | N | Y | Y |
| RCA4 | MPI-ESM-LR | Y | Y | Y |
| RCA4 | CSIRO-Mk3-6-0 | N | Y | Y |
| REMO2009 | MIROC5 | Y | N | N |
| REMO2009 | GFDL-ESM2G | Y | N | N |
| REMO2009 | CM5A-LR | Y | N | Y |
| REMO2009 | HadGEM2-ES | Y | N | N |
| REMO2009 | EC-EARTH | N | Y | Y |
| REMO2009 | MPI-ESM-LR | Y | Y | Y |





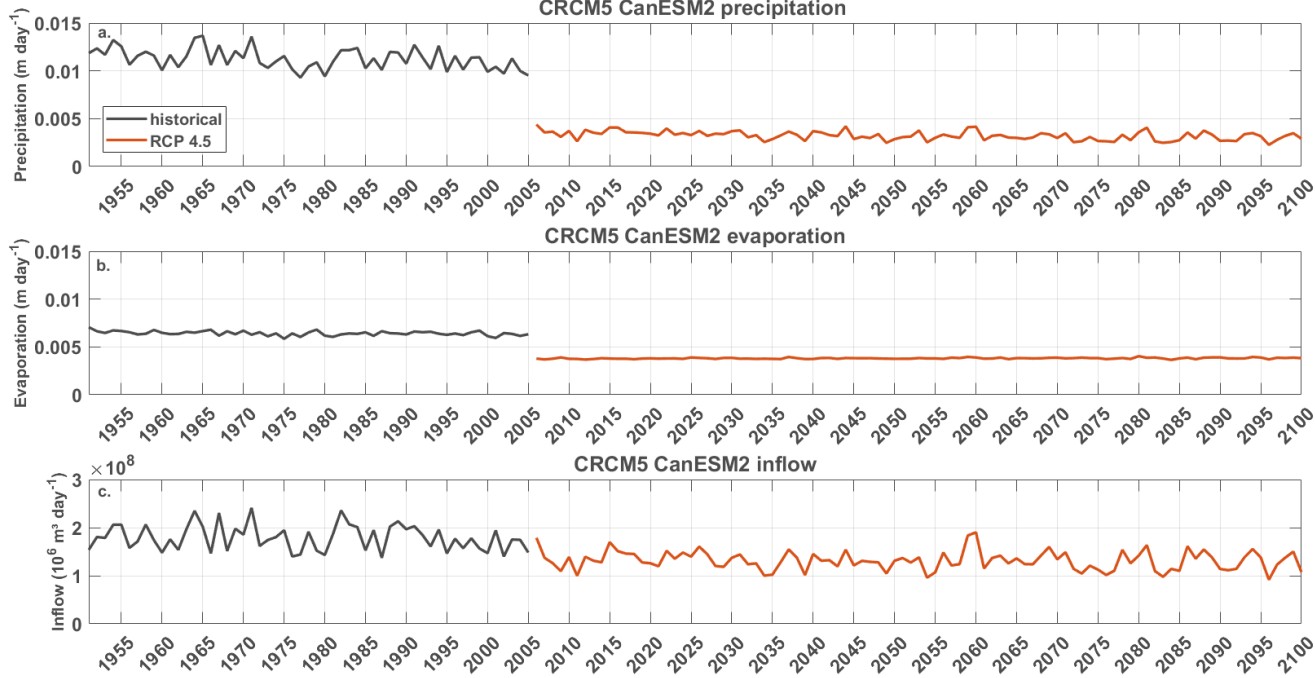

**Figure A2.** WB terms following historical and RCP 4.5 of CRCM5-CanESM2 simulations (without bias correction).



## Appendix B:  Correction of CORDEX ensemble members for number of days

Not all simulations from the CORDEX ensemble have the same number of days. As a fixed number of days is a necessary condition to compared the WBM simulations, a correction was applied on the daily WB terms of some simulations.

5      First, simulations driven by HadGEM2-ES (CCLM4-8-17, RACMO22T, RCA4, REMO2009), have 30 day-months and only go until 2099. To account for the missing days, 5 extra days are added for every 72 days in the year, starting after the 36th day. The added days are the average of the respective WB terms during the previous and next day. In addition, we accounted for the fact that these model simulations do not include the year 2100, by repeating the year 2099. The simulations with HadGEM2-ES for RCP 4.5 have no december month in the year 2099. This is also the case for the HadGEM2-ES CCLM4-8-17 simulation
10    for RCP 8.5. In both cases, December 2099 is added by repeating the month November of the same year.

       Finally, in all simulations that do not account for leap years (RCMs driven by CanESM2, NorESM1-M, MIROC5, GFDL-ESM2M, CM5A-MR and CSIRO-Mk3-6-0), an extra day in the leap years is added by taking the average WB term value of the days corresponding to the 28th of February and the 1st of March. Overall, compared to the total number of days of the
15    future projections (34698 days), we note that corrections on single days (up to 888 days depending on the simulations) have a little influence on the outcomes of this study.

## Appendix C:  Simulations with empirical quantiles bias correction

When a bias correction based on empirical quantiles is used, very similar results are found (compare figs. C1, C2, C3 and C4). Based on this, we conclude that the bias correction methods has very little effect on the results presented in this study.





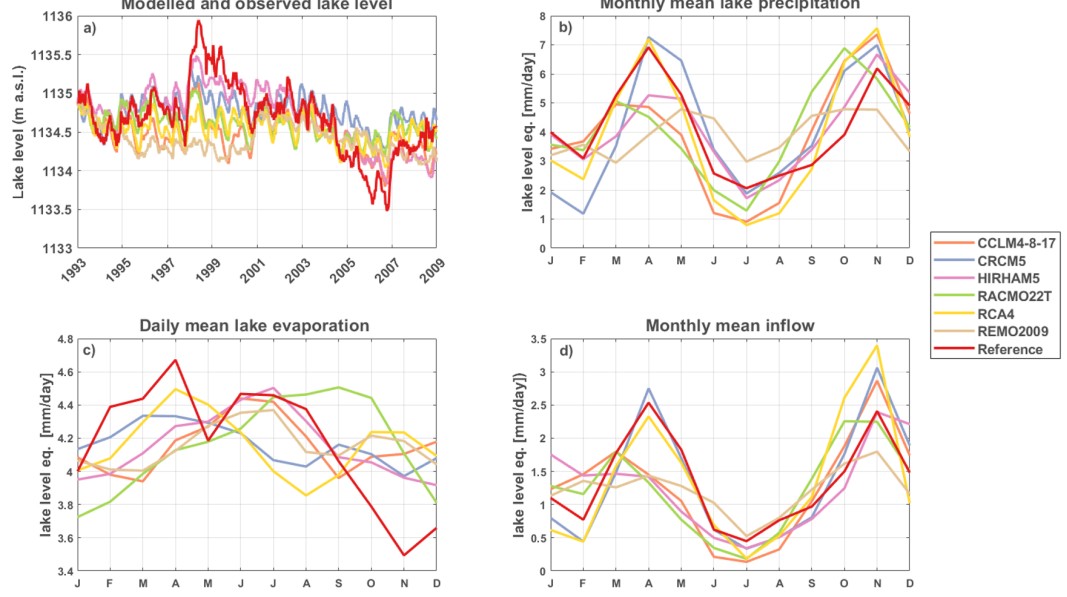

**Figure C1.** As in fig. 5, but bias corrected using empirical quantiles.

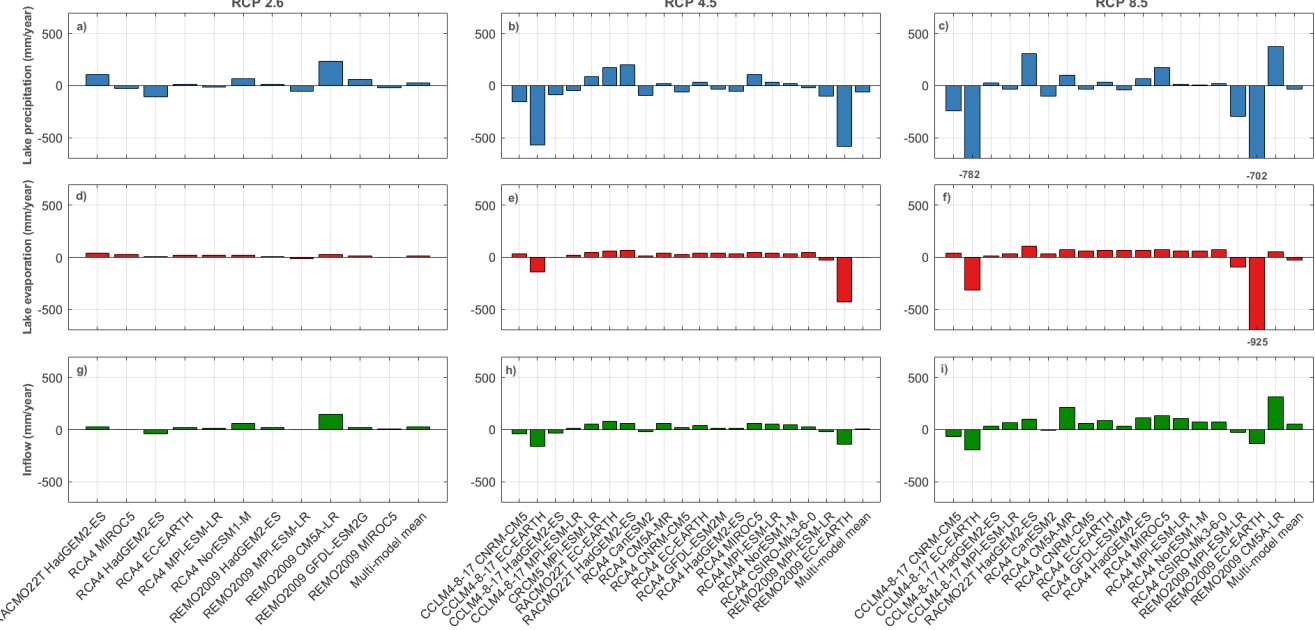

**Figure C2.** As in fig. 7, but bias corrected using empirical quantiles.





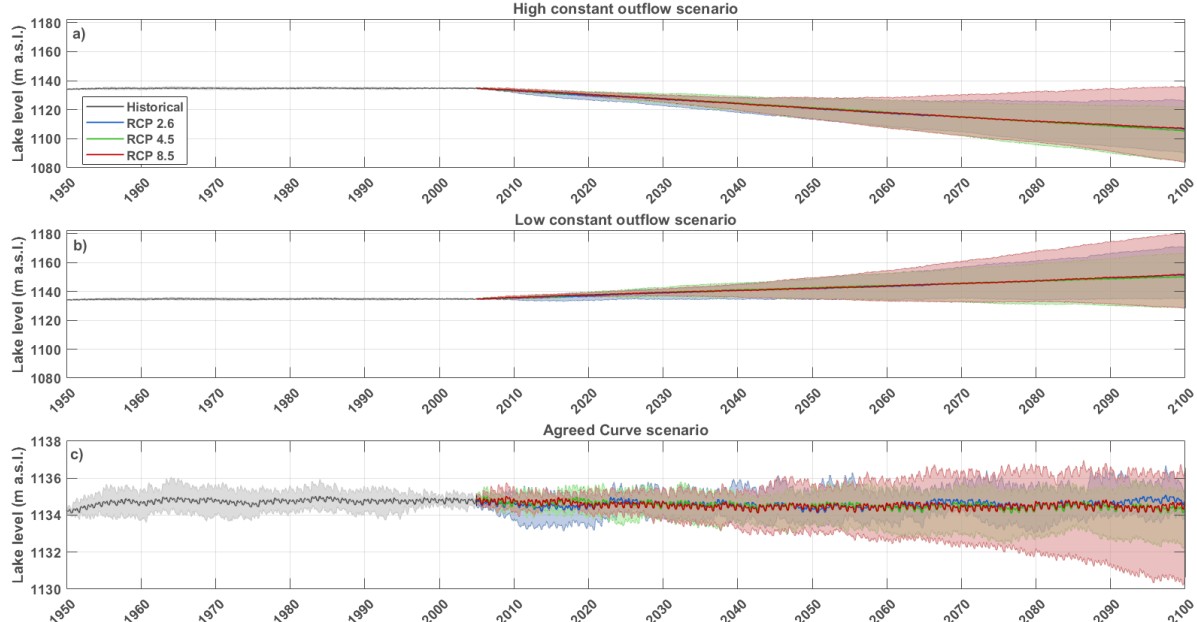

**Figure C3.** As in fig. 8, but bias corrected using empirical quantiles.

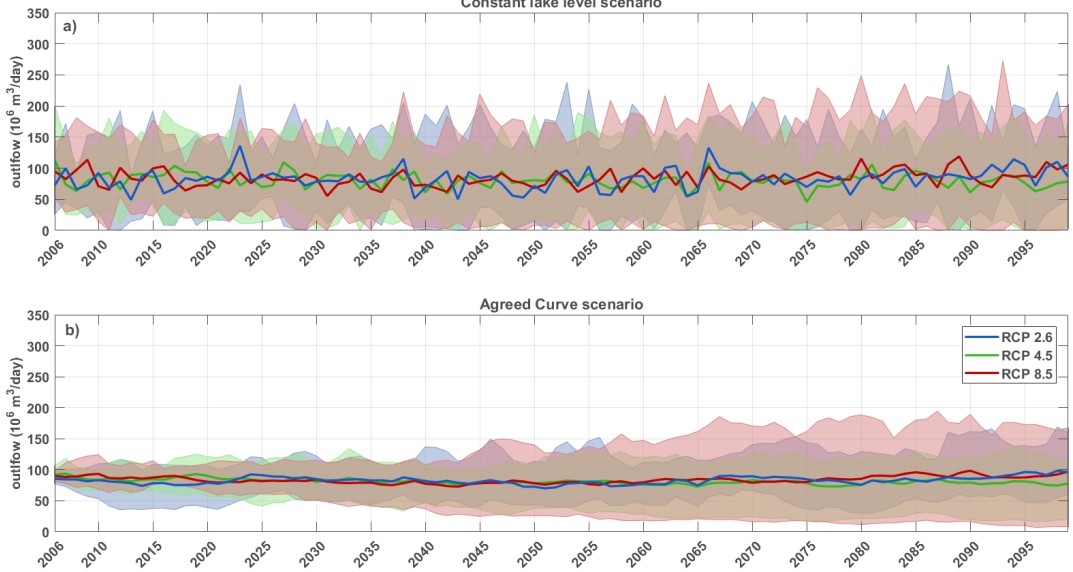

**Figure C4.** As in fig. 9, but bias corrected using empirical quantiles.

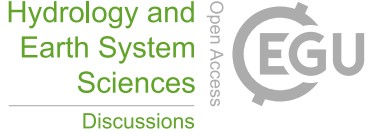

*Competing interests.* The authors declare that they have no conflict of interest.

*Acknowledgements.* We acknowledge the CLM-community (clm-community.eu) for developing COSMO CLM$^2$ and making the model code available. Computational resources and services used for the COSMO-CLM$^2$ simulations were provided by the VSC (Flemish Supercomputer Center), funded by the Hercules Foundation and the Flemish Government department EWI. In addition, we are grateful to the World Climate

5   Research Programme (WRCP) for initiating and coordinating the CORDEX-Africa initiative, to the modelling centres for making their downscaling results publicly available through ESGF.Wim Thiery was supported by an ETH Zurich postdoctoral fellowship (Fel-45 15-1). The Uniscientia Foundation and the ETH Zurich Foundation are thanked for their support to this research.



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
