# Peer review of "Modelling the water balance of Lake Victoria (East Africa), part 2: future projections"

_Hydrology and Earth System Sciences, 2018_

## Referee Comment (RC1) · Anonymous Referee #1 · 24 Apr 2018

GENERAL COMMENTS:

The authors write about impacts of climate change on the Water Balance of the Lake Victoria. The authors are apparently with a climate background and are using terms that for a readership with hydrology/water resources management background are somewhat unusual. The authors use the term "outflow scenario", however, the outflow is the result of the management/operation of the dams at the outlet. Therefore I recommend to call these scenarios "management OR operation scenarios". I also suggest replacing "amount(s)" (e.g. "amount of outflow") by "volume(s)".

Furthermore, there are some relevant points that need clarification (and I don't find an answer in this neither in the first paper): when following the Agreed Curve in the simulations (using RCM results as input, evaluation and climate scenarios), where are

minimum and maximum water levels set for this curve (1133.5 and 1136.0 (1136.5?) m asl, respectively)? In case the water level is lower there is no outflow (but further reduction of water level due to evaporation possible). What happens if the water level is reaching or surpassing the maximum water level? All water is discharged?

SPECIFIC COMMENTS:

Abstract:

Page 1; Line 4: "Yet, nothing is known about..." change to "Yet, little is known about..."

1. Introduction:

Page 1; Lines 9, 11, 13: Replace "emission" by "concentration"

Page 1; Line 24: "...during the dry season in the Ethiopian highlands (Di Baldassarre et al., 2011)." change to "...during the dry season in the Ethiopian highlands, where the second major source is located (Di Baldassarre et al., 2011)."

Page 2; Lines 1, 3, 6 (and whole text): use either "Nile basin" or "Nile Basin"

Page 2; Line 4: "...hydropower." change to "...hydropower generation."

Page 2; Lines 6/7: "Lake level fluctuations also influence the amount of outflow released..." it is clear that input (precipitation over the lake and inflow by tributaries) affects the lake level, but also the (managed) outflow has a strong influence on the lake level. I would rather see that "The amount of outflow released also influence the lake level fluctuations...".

Page 2; Line 13: "...relating the outflow and lake level" would recommend changing to "...relating the lake level and outflow" as lake level is the guiding dimension

Page 2; Lines 19/20: "In the last decades, the long rain seasons in East Africa have experienced a series of droughts..."; this statement is somewhat ambiguous to me: with "long rain seasons" you mean the annually occurring rain seasons (with peaks in

March-May and Nov-Dec) – in contrast to the (annual) dry season – that have shown low precipitation volumes (sums) in the last decades? Maybe rephrase to "In the last decades, the usually long and strong rain seasons in East Africa have shown a number of years with precipitation volumes below average…" to make this statement more clear.

Page 3; Line 15: "…CLimate…" change to "…Climate…"

Page 3; Line 19: Replace "emission" by "concentration"

Page 3; Line 21 (and whole text): "Vanderkelen et al., 2018a" there is only one reference for Vanderkelen et al. in 2018.

Page 3; Line 24: please delete "successfully"

2. Data and methodology

Page 3; Line 31: Change to "…the Representative Concentration Pathways (RCP) 2.6,…" AND then change Page 4; Line 2 "...the three Representative Concentration Pathways (RCPs; 2006-2100)." to "...the three RCPs (2006-2100)."

Page 3; Line 31: Replace "emission" by "concentration"

Page 5; Line 6: "...with the observed lake level in 1950" guess you mean "1951".

Page 5; Line 9: "The evaluation and historical WBM simulations use recorded outflow values." As far as I understand you use the recorded outflow values (shown in Fig. 3 of the first paper) in the historical WBM simulations using the input (precipitation and evaporation over the lake and inflow by tributaries) from the (RCMs) historical climate simulations. As the outflow is depending on the lake level it clearly depends on weather conditions (precipitation, inflow) of the last weeks/months, e.g. rain/dry season with corresponding high/low lake levels. You also stated that there are huge differences in the climate between the years, affecting precipitation over the lake and inflow by tributaries. As a day/month/year in a RCM is representing the general climatic

characteristics of that period, but not the weather of this specific day/month/year, it is not clear why you do not use the Agreed Curve for calculating the outflow. It is clear to me that the way the simulations are done only by chance you can close the WB of the lake. When a RCM simulates a sequence of wet years (with consequently high outflow volumes) while a sequence of dry years (with consequently low outflow volumes) was observed, using the low observed outflow in the RCM simulation will inevitably lead to problems in closing the WB.

Page 5; Lines 16/17: Here it is explained for the "constant lake level scenario": "If the water balance is negative, there is no outflow, but the lake level is allowed to decrease. When the water balance is positive again, the lake level is first restored to its predefined constant height." It is not mentioned how the outflow at very high water levels, i.e. higher than maximum water level of 1136.0 (or 1136.5) m asl, is simulated.

Page 5; Lines 19/20: "In the last defined scenarios, future outflow is kept constant, ensuring a constant supply of hydropower" - as stated by the authors the water level fluctuates up to 2.5 m (e.g. page 13, line 24 of the first paper). The same outflow with a difference in the head of 2.5 m will lead to differences in the generated hydropower. There may be a "constant supply of hydropower" but as this could be mistaken as "monotonic" or "unchanged" I think this should be rephrased (maybe use "continuous"?).

Page 6; Line 9: "...of the 7 tested the bias..." change to "...of the 7 tested bias..."

3. Results:

Page 6; Line 26: "Results of the evaluation WBM run are compared..." change to "Results of the evaluation WBM runs are compared..."

Page 7; Line 1: "Table 1. Table showing the a and b parameters of the linear..." change to "Parameters a and b of the linear..."

Page 8; Line 2: "...lake levels (figure 4a)." change to "...lake levels (Fig. 4a)."

Page 9; Line 9ff: the whole section (3.3), giving the results for water levels and outflow, needs to be revised. According to my understanding water levels above 1136.5 m asl would lead to increasingly high outflows (there is no structure to withhold water) and the water level cannot reach values as given, e.g., in Fig. 8 (=> reaching up to 1180 m asl). It is not clear what (if any) digital elevation model is used. A water level of 1180 m asl would mean (considering the flat shores/wetlands close to the lake) that an enormous area (lake surface area > 100.000 km2?) would be flooded.

Page 9; Line 26: "...relation between outflow and lake level" would recommend changing to "...relation between lake level and outflow" as lake level is the guiding dimension

4. Discussion:

Page 10; Lines 12/13: "...LVB (Fig 3 and Fig. 4)." change to "...LVB (Figs. 3 and 4)."

Page 10; Lines 21/22: "The other CORDEX models have no lake model embedded." – Why this information is only given in the discussion? It would help to understand better the deviations of the other models when giving already in "2. Data and methodology".

Page 10; Line 33: "...Souverijns et al. (2016) (up to 0.5 mm-1 over the lake)." change to "...Souverijns et al. (2016) (up to 0.5 mm day-1) over the lake."

Page 12; Line 20: "Providing a constant hydropower supply, which implies a constant outflow,..." - The same outflow with a difference in the head of 2.5 m will lead to differences in the generated hydropower...

Page 12; Line 22ff: Please check where you are referring to (or citing numbers for) electricity generation/demand and energy demand (energy also includes fuel for cars etc., the energy demand of a region/country therefore is higher than the electricity demand).

Page 13; Lines 6 and 15: "...model uncertainty..." - Here you are referring to CLIMATE model uncertainty?

[Figure]

5. Conclusion:

Page 13; Line 25: "...a water balance model constructed developed for Lake Victoria..." change to "...a water balance model developed for Lake Victoria..."

Appendix A (pages 23-25):

"...HIRHAM5 driven by EC-EARTH and CRCM5 driven by CanESM2 are not used due to the fact that they exhibit discrepancies between their historical and future simulations..." – please check if during the download, naming of files or at some other point (even check if an error occurred at the CORDEX homepage) an error occurred: According to Fig. A1 for the historical run (HIRHAM5) the precipitation is between 3-5 mm/day, while for scenarios (first years) according to Fig. A2 CRCM5 gives comparable values. According to Fig. A2 for the historical run (CRCM5) the precipitation is between 10-14 mm/day, while for scenarios (first years) according to Fig. A1 HIRHAM5 gives comparable values. Can it be that historical runs or scenarios between these two models were substituted (I see the same for evaporation and to a lesser extend for inflow)?

Appendix B (page 26):

Line 3: "...condition to compared the WBM simulations..." change to "...condition to compare the WBM simulations..."

Line 6: "To account for the missing days, 5 extra days are added for every 72 days in the year, starting after the 36th day." – please describe exactly how this is done; I guess that "1 extra day is added each 72 day, starting at the 36th day." as shown in the table below

day of year (orig.) 36 108 180 252 324 360

day of year (corr.) 37 110 183 256 329 365

Appendix C (pages 26-28):

Line 18: "...compare figs. C1, ..." change to "...compare Figs. C1, ...", also in the titles of Figs. C1, C2, C3 and C4 "As in Fig. ..."

Acknowledgements (page 28):

Line 6: "...through ESGF.Wim Thiery was..." change to "...through ESGF. Wim Thiery was..."

Please check the References, e.g.:

Page 30; Line 2: "Adeyemi, K. O. and Asere, A. A.: a Review..." change to "Adeyemi, K. O. and Asere, A. A.: A Review..."

Page 31; Lines 20/21: "Mayaux, P., Massart, M., Cutsem, C. V., Cabral, a., Nonguierma, a., Diallo, O., Pretorius, C., Thompson, M., Cherlet, M., Defourny, P., Vasconcelos, M., Gregorio, a. D., Grandi, G. D., and Belward, a.:..." change to " Mayaux, P., Massart, M., Cutsem, C. V., Cabral, A., Nonguierma, A., Diallo, O., Pretorius, C., Thompson, M., Cherlet, M., Defourny, P., Vasconcelos, M., Gregorio, A. D., Grandi, G. D., and Belward, A.:..."

Page 32; Lines 23/24: is this a book chapter (editor, publisher?) or a journal article?

Page 33; Lines 1/2: "Vanderkelen, I., Lipzig, N. P. M. V., and Thiery, W.: Modelling the water balance of Lake Victoria (East Africa), part 1 : observational" change to "Vanderkelen, I., Lipzig, N. P. M. V., and Thiery, W.: Modelling the water balance of Lake Victoria (East Africa), part 1: observational" and include full reference (page no. etc.)

---

## Referee Comment (RC2) · Anonymous Referee #2 · 9 Jul 2018

General comments:

The paper presents an interesting study, very relevant from the point of view of future management of the Lake Victoria system. Messages of relevance are formulated in the paper. However, it is relatively difficult to discover these messages, as the paper unnecessarily focuses on the process of arriving at them rather than on the messages themselves. The focus on the process is not really justified, as no new approaches are used, nor methodologies are developed. The value of the paper is in the messages, description of methodology is necessary only to the extent that makes the messages defensible.

There is an important consequence to the above. It is well known that GCMs and RCMs have biases. The purpose of the paper is not to evaluate performance of a set of RCMs

in the Victoria Lake Basin, but to assess impacts of future climate on Lake Victoria's water balance. A large section of the paper, however, deals with RCM biases and evaluation of RCM performance in LVB. In my opinion, that section should be presented only if the results of evaluation were used to select a subset of models/simulations to be used in further analyses.

Bias correction based on 12 years of data is not appropriate, unless the authors are able to show that in the studied region decadal and multidecadal variability is very low.

The reason to consider the four outflow scenarios should be described better. They are not compatible with each other in that they represent totally different management approaches rather than a quantitative modification of a particular approach. It would make sense to present them if the authors were targeting a question of most efficient or effective or robust management approach under changing climate. But they are not. The justification of using the four is actually very weak and conceptually incorrect. The fact that in the past Agreed Curve was not adhered to does not mean that the other three are preferred alternatives.

Specific comments: 1)PERSIANN rainfall, and COSMO-CLM are hardly what one would understand as "observations". Aren't there "real" observational data available? Rain gauges? Met stations? if not - then why these products were chosen? why not CHIRPS? or TRMM/GPM?

2) there is no need to show the HIRHAM discrepancy in the appendix. It's enough if it is stated clearly in the text. 3) what is the outer polygon in Figure 1? NB. a study area map showing the lake, main rivers, catchment and location of dams would be beneficial. 4) fig 6: in the caption, climate change signal is described as a difference between historical and future. In such a case, is change with positive sign an increase into the future? or a decrease into the future? 5) fig 6 and fig 7 show essentially the same information. Only fig 7 could be used. Bias correction could change signal, but only under very specific conditions. Also, showing 95% confidence interval would help

assessing the strength of signal. 6) Changes in precipitation and evaporation should be expressed as a percent (relative change), not as a difference (absolute change). Change expressed as difference in a situation of strong biases (offsets) may result in obtaining negative values when that change is compared to observations. 7) Table 1 is not necessary, and values of coefficient "a" shown in it seem unrealistic. If model has shown no bias, value of a would be 0. Table 1 has values in the order of 10000000. 8) what is Pm in eq. 3? 9) pg. 13, line 11: uncertainties as represented by the spread of the analysed ensemble do not result just from model deficiencies, but are also due to initial condition uncertainty.

---

## Author Comment (AC1) · 28 Sep 2018

**Modelling the water balance of Lake Victoria (East Africa), part 2: future projections**

Hydrology and Earth System Sciences

September 28, 2018

Inne Vanderkelen[1], Nicole P. M. van Lipzig[2], Wim Thiery[1, 3]
inne.vanderkelen@vub.be

[1] Department of Hydrology and Hydraulic Engineering, Vrije Universiteit Brussel, Pleinlaan 2, 1050 Brussels, Belgium.

[2] Department of Earth and Environmental Sciences, KU Leuven, Celestijnenlaan 200E, 3001 Leuven, Belgium.

[3] Institute for Atmospheric and Climate Science, ETH Zurich, Zurich, Switzerland.

**Contents**

**Abstract**

This response letter contains numbered figures and references to these figures. To prevent confusion, the figures embedded within this response letter are called illustrations. Finally, the following convention is applied to denote modification in the original manuscript: new text.

**1   Reviewer 1**

> ### Reviewer 1 Comment 1
>
> The authors write about impacts of climate change on the Water Balance of the Lake Victoria. The authors are apparently with a climate background and are using terms that for a readership with hydrology/water resources management background are somewhat unusual. The authors use the term "outflow scenario", however, the outflow is the result of the management/operation of the dams at the outlet. Therefore I recommend to call these scenarios "management OR operation scenarios". I also suggest replacing "amount(s)" (e.g. "amount of outflow") by "volume(s)"

**Response**

We thank Reviewer 1 for his/her thorough review of the study with plenty suggestions to improve the manuscript. Below, we address every comment carefully and explain the corresponding changes in the manuscript.

Concerning the first issue about the terms used for operation or management scenario, we agree with the reviewer that the outflow at the dam results directly from the 'management' or 'operation' scenario, consistent with the terms used in literature. We therefore replaced the term 'outflow scenario' by '(dam) management scenario' throughout the manuscript. We also followed the suggestion to replace 'outflow amounts' by 'outflow volumes', and updated the manuscript accordingly.

> ### Reviewer 1 Comment 2
>
> Furthermore, there are some relevant points that need clarification (and I do not find an answer in this neither in the first paper): when following the Agreed Curve in the simulations (using RCM results as input, evaluation and climate scenarios), where are minimum and maximum water levels set for this curve (1133.5 and 1136.0 (1136.5?) m asl, respectively)? In case the water level is lower there is no outflow (but further reduction of water level due to evaporation possible). What happens if the water level is reaching or surpassing the maximum water level? All water is discharged?

**Response**

We thank the reviewer for raising this comment. The Agreed Curve is calibrated and calculated based on a study of gaugings at Namasagali, before the construction of the dam (Sutcliffe and Parks, 1999; Sene, 2000). In 1961-1964, the lake level rose above the limit of the gaugings on which the Agreed Curve was based, and the Agreed Curve was extrapolated to account for this extra outflow. Since then, historical lake levels have been fluctuating between the observed range for which the Agreed Curve is known, between 10 and 13.5 m, as measured at the dam in Jinja (Sutcliffe and Petersen, 2007). These values correspond to the absolute heights of 1133 and 1136.5 m, converted by using the EIGEN-6c3stat geoid model (Vanderkelen et al., 2018; Schwatke et al., 2015).

The lake levels as a result of the RCM evaluation simulations use the recorded outflow values, and do not follow the Agreed Curve directly. The outflow of the historical and future RCM simulations is calculated based on the Agreed Curve. In the original version of manuscript, we extrapolated the relationship used by Sene (2000, Eq. 1) to calculate outflow

volumes for lake levels surpassing the historical range of 1133 to 1136.5 m. We however agree with the reviewer to impose physical constraints to the lake level fluctuations following the Agreed Curve. These limits correspond with the observed lake level range, 1133 to 1136.5 m.

Likewise we updated the manuscript, first we added the range in the description of the Agreed Curve in the introduction:

> Since the construction of the dam complex in 1954, a rating curve called the "Agreed Curve" was established relating the lake level and outflow in natural conditions (Sene, 2000):
>
> $$Q_{out} = 66.3(L - 7.96)^{2.01} \tag{1}$$
>
> In this equation, the dam outflow $Q_{out}$ (m$^3$ day$^{-1}$) is calculated based on the lake level, $L$ (m), as directly measured at the dam. The Agreed Curve can be used to calculate outflow volumes based on lake levels which lie in the historical observed range going from 10 to 13.5 measured at the dam (Sutcliffe and Parks, 1999).

Second, we updated the description of the Agreed Curve management scenario (paragraph 2.3):

> A first assumption is that future outflow starts following the Agreed Curve again. In this *Agreed Curve scenario*, daily outflow is calculated following Eq. 1 based on the lake level of the previous day. Lake level fluctuations are restricted to fluctuate within the historical observed range (10 m to 13.5 m measured at the dam, corresponding to 1130 m to 1136.5 m a.s.l), as this is the range for which the Agreed Curve is known. If the lake level of the previous day drops below the lower limit, the outflow is set to 0 m$^3$ day$^{-1}$ and if the lake level rises above the upper limit, all additional water is discharged.

**Reviewer 1 Comment 3**

P1, L4: "Yet, nothing is known about..." change to "Yet, little is known about..."

**Response**

We adjusted the manuscript.

**Reviewer 1 Comment 4**

P1, L24: ". . .during the dry season in the Ethiopian highlands (Di Baldassarre et al., 2011)." change to ". . .during the dry season in the Ethiopian highlands, where the second major source is located (Di Baldassarre et al., 2011)."

**Response**

We adjusted the text.

> **Reviewer 1 Comment 5**
>
> P2, L1,3,6: (and whole text): use either "Nile basin" or "Nile Basin"

**Response**

Now, "Nile Basin" is used throughout the whole manuscript.

> **Reviewer 1 Comment 6**
>
> P2 L4: ". . .hydropower." change to ". . .hydropower generation."

**Response**

We followed the suggestion and updated the text.

> **Reviewer 1 Comment 7**
>
> P2 L6-7: "Lake level fluctuations also influence the amount of outflow released..." it is clear that input (precipitation over the lake and inflow by tributaries) affects the lake level, but also the (managed) outflow has a strong influence on the lake level. I would rather see that "The amount of outflow released also influence the lake level fluctuations...".

**Response**

We understand the comment of the reviewer. We agree that the outflow has an influence on the lake level as well. But in the current (and historical) dam management regime following the Agreed Curve, the outflow volume is based on the lake level, which is the result in his turn of the combined effect of lake precipitation, lake evaporation and inflow. Therefore, we adjusted the sentence highlighting the two way-interaction between lake level and outflow.

> While the released outflow affects the lake level, climate-driven lake level fluctuations also influence the outflow volume released by the dam. The hydropower potential, and thus energy availability in the region therefore strongly depends on the interplay between outflow and lake levels. Information on the future evolution of the levels and outflow volumes of Lake Victoria is therefore vital for future generations living on its coasts.

> **Reviewer 1 Comment 8**
>
> P2 L13: ". . .relating the outflow and lake level" would recommend changing to ". . .relating the lake level and outflow" as lake level is the guiding dimension

**Response**

We applied the comment and updated the manuscript.

**Reviewer 1 Comment 9**

P2 L19-20: "In the last decades, the long rain seasons in East Africa have experienced a series of droughts. . ."; this statement is somewhat ambiguous to me: with "long rain seasons" you mean the annually occurring rain seasons (with peaks in March-May and Nov-Dec), in contrast to the (annual) dry season, that have shown low precipitation volumes (sums) in the last decades? Maybe rephrase to "In the last decades, the usually long and strong rain seasons in East Africa have shown a number of years with precipitation volumes below average. . ." to make this statement more clear.

**Response**

We understand the confusion of the reviewer, and we clarified this point by adding information about the short and long rain season in the Lake Victoria Basin:

> The region is a hotspot for climate change, as it is very likely that climate change will have a major influence on precipitation (Nicholson, 2017; Kent et al., 2015; Otieno and Anyah, 2013; Souverijns et al., 2016). Precipitation over the Lake Victoria Basin (LVB) knows a seasonal cycle with two main rainfall seasons: the long rains in March, April and May and the short rains in September, October and November (Williams et al., 2015). In the last decades, the long rain seasons in East Africa have experienced a series of droughts (Lyon and Dewitt, 2012; Rowell et al., 2015; Souverijns et al., 2016; Nicholson, 2016, 2017), while there was no trend observed for the short rains due to a large year-to-year variability (Rowell et al., 2015). This drying trend of the long rains is in contrast with climate model projections for the upcoming decades, projecting an increase in precipitation over East Africa (Otieno and Anyah, 2013; Kent et al., 2015).

**Reviewer 1 Comment 10**

P3 L15: ". . .CLimate. . ." change to ". . .Climate. . ."

**Response**

We adjusted the manuscript accordingly.

**Reviewer 1 Comment 11**

P3 L19: Replace "emission" by "concentration"

**Response**

We replaced "emission pathways" by "concentration pathways".

**Reviewer 1 Comment 12**

P3 L21 an whole text :Vanderkelen et al., 2018a" there is only one reference for Vanderkelen et al. in 2018.

**Response**

The "Vanderkelen et al., 2018a" refers to the first paper of the two-part paper series. The first paper is accepted for publication in the journal Hydrology and Earth System Sciences, but will be published together with this paper, as a accompanying paper. In this way, the references to both papers will be consistent. Both will be referred as "Vanderkelen et al., 2018". This is adjusted in the manuscript.

> **Reviewer 1 Comment 13**
>
> P3 L24: please delete "successfully"

**Response**

We removed it in the sentence.

> **Reviewer 1 Comment 14**
>
> P3 L31: Change to ". . .the Representative Concentration Pathways (RCP) 2.6,..." AND then change Page 4; Line 2 "...the three Representative Concentration Pathways (RCPs; 2006-2100)." to "...the three RCPs (2006-2100)."

**Response**

We thank the reviewer for the correction of the acronym introduction. We rephrased the sentences to make the introduction of the acronyms more clear.

> In recent years, RCM downscalings of the Coupled Model Intercomparison Project Phase 5 (CMIP5) GCMs have become available through the CORDEX framework. The CORDEX-Africa project provides simulations over the Africa domain, which includes the whole African continent with a spatial resolution of 0.44° by 0.44° and a daily output frequency (Nikulin et al., 2012). In CORDEX-Africa, there are currently simulations with six RCMs available (*CCLM4-8-17, CRCM5, HIRHAM5, RACMO22T, RCA4* and *REMO2009*) for the historical (1950-2005) and future period (2006-2100) under Representative Concentration Pathways (RCPs) 2.6, 4.5 and 8.5.

> **Reviewer 1 Comment 15**
>
> P3 L31: Replace "emission" by "concentration"

**Response**

We updated the manuscript.

> **Reviewer 1 Comment 16**
>
> P5 L6: "...with the observed lake level in 1950" guess you mean "1951".

**Response**

We thank the reviewer for raising this point. The CORDEX simulations start indeed in 1951, as stated a few sentences before. We updated this in the text, as well as the starting lake

level for both the evaluation and historical simulations, which were incorrect values. We apologize for this error.

> The evaluation simulations start with the observed lake level in 1990 (1135 m a.s.l.) and the historical simulations with the observed lake level in 1951 (1133.7 m a.s.l.).

Accordingly, we adjusted the x-axis labeling of Fig. 8 in the original manuscript and Fig. C3, so that the correct period (1951-2100) is displayed (illustration 2).
* * *
**Reviewer 1 Comment 17**

P5 L9: "The evaluation and historical WBM simulations use recorded outflow values." As far as I understand you use the recorded outflow values (shown in Fig. 3 of the first paper) in the historical WBM simulations using the input (precipitation and evaporation over the lake and inflow by tributaries) from the (RCMs) historical climate simulations. As the outflow is depending on the lake level it clearly depends on weather conditions (precipitation, inflow) of the last weeks/months, e.g. rain/dry season with corresponding high/low lake levels. You also stated that there are huge differences in the climate between the years, affecting precipitation over the lake and inflow by tributaries. As a day/month/year in a RCM is representing the general climatic characteristics of that period, but not the weather of this specific day/month/year, it is not clear why you do not use the Agreed Curve for calculating the outflow. It is clear to me that the way the simulations are done only by chance you can close the WB of the lake. When a RCM simulates a sequence of wet years (with consequently high outflow volumes) while a sequence of dry years (with consequently low outflow volumes) was observed, using the low observed outflow in the RCM simulation will inevitably lead to problems in closing the WB.
* * *
**Response**

We completely agree with the reviewer on this issue and followed this correct approach in the manuscript: (i) for the evaluation simulations, based on the RCM reanalysis downscalings, the recorded outflow values are used, whereas (ii) in the historical simulations, based on the GCM downscalings, historical lake levels calculated based on the Agreed Curve for each RCM-GCM combination. The figures in the manuscript are therefore correct. However, in the text, this was described wrongly and slipped out of our attention. We apologize for this inconsistency and updated the text accordingly:

> The evaluation WBM simulations use recorded outflow values. In the historical WBM simulations, the outflow is calculated based on the Agreed Curve. While observed outflow volumes are available for the historical period, these cannot be used in the WBM as RCMs driven by GCMs represent the general climatology and do not account for the actual observed weather, reflected in the recorded outflow volumes.

> ### Reviewer 1 Comment 18
>
> P5 L16-17: Here it is explained for the "constant lake level scenario": "If the water balance is negative, there is no outflow, but the lake level is allowed to decrease. When the water balance is positive again, the lake level is first restored to its predefined constant height." It is not mentioned how the outflow at very high water levels, i.e. higher than maximum water level of 1136.0 (or 1136.5) m asl, is simulated.

**Response**

So far we did not account for outflow at very high water levels. However, by construction, very high water levels are never reached in the *constant lake level scenario*, as the outflow is calculated each day based on the residual of the water balance of the previous day . As a result, the lake level will never be higher than the highest constant (1335.2 m; illustration 1). In a very wet period, with a lot of lake precipitation and inflow, the extra water volumes will lead to an increased outflow, and not to an increased lake level, by construction. Here, it is possible that the physically maximum outflow of the dam is reached and the excess water would overflow the dam. In this case, the excess water would still be released.

However, for completeness we constrained future lake level variations in the revised manuscript for all dam management scenarios. See next comment for the updated manuscript text.

[Figure]

Illustration 1: Lake level projections following the constant lake level scenario.

> **Reviewer 1 Comment 19**
>
> P5 L19-20: "In the last defined scenarios, future outflow is kept constant, ensuring a constant supply of hydropower" - as stated by the authors the water level fluctuates up to 2.5 m (e.g. page 13, line 24 of the first paper). The same outflow with a difference in the head of 2.5 m will lead to differences in the generated hydropower. There may be a "constant supply of hydropower" but as this could be mistaken as "monotonic" or "unchanged" I think this should be rephrased (maybe use "continuous"?).

**Response**

We thank the reviewer for this comment, as we did not take this into account. The amount of hydropower produced is proportional to both the outflow and the water head. Therefore, a constant outflow does indeed not imply a constant hydropower production. The unrealistic increase in lake levels as represented by the original version of the manuscript in fig. 8b, would lead to an increase in hydropower. Therefore, we decided to make the dam management scenarios more realistic by changing them from managing a constant outflow volume to managing for a constant hydropower production. Likewise, we updated the manuscript in different sections.

The dam management section in the methods (section 2.3):

> In the last defined scenarios, future outflow is regulated in order to provide a constant hydropower production without interruptions while lake levels fluctuate in their physical range. In the study of Koch et al. (2013), hydropower production of a reservoir is quantified as

$$P_{el} = Q_{out} \times h \times k \qquad (2)$$

> with $Q_{out}$ the outflow of the reservoir in m$^3$ day$^{-1}$, $P_{el}$ the electricity produced (kW), $h$ the water head (m) and $k$ the efficiency factor (kN/m$^3$). After rearranging this equation and adding a constraint to maximum outflow, the outflow needed to produce a firm amount of electricity is given by

$$Q_{out} = MIN(\frac{P_{el}}{h \times k}, Cap_{hpp}) \qquad (3)$$

> with $Cap_{hpp}$ the maximum turbine flow capacity (m$^3$/s). Lake Victoria is controlled by both the Nalubaale and Kiira dams. As these dams operate parallel of each other, we simplified the analysis by assuming only one dam regulating the outflow, with the combined hydropower capacity of both real dams. This results in a $Cap_{hpp}$ of 1150 m$^3$/s (the average of 1200 m$^3$/s for Nalubaale and 1100 m$^3$/s for Kiira; Kizza and Mugume, 2006). $h$ is assumed to be equal to the relative lake level, as measured at the dam. For the efficiency factor $k$ a value of 13.77 kN/m$^3$ is used, calculated from eq. 2 using the values for maximum turbine flow $Cap_{hpp}$, maximum water head ($h_{max}$ = 24 m) and the sum of maximum electricity production, 380 MW (180 MW for Nalubaale and 200 MW for the Kiira dam; Kull, 2006). Based on this, two management scenarios providing a constant hydropower production (HPP) are defined. The *historical HPP management scenario* prescribes a daily HPP equal to the mean historical HPP ($P_{el}$ = 168

MW), which is calculated using eq. 2 with the historical observed mean outflow ($88 \cdot 10^6$ m$^3$/day) and the mean relative lake level (11,9 m). Second, in the *high HPP management scenario*, HPP is set equal to the electricity produced in the year in which the outflow was maximum ($P_{el}$ = 247 MW, in 1964 with an outflow of $138 \cdot 10^6$ m$^3$ day$^{-1}$).

In both the constant lake level scenario and the HPP management scenarios, we impose physical constraints to the lake level fluctuations: the lake level should fluctuate between 0 m and 26 m as measured at the dam, 1122.9 m and 1146.9 m a.s.l. (the height of the dam is 31 m with a safety level of 7 m). Similar to the limits of the Agreed Curve, the outflow is set to 0 m$^3$ day$^{-1}$ whenever the lake level drops below 1122.9 m a.s.l. and all additional water is discharged by the sluice gates if the lake level rises above 1146.9 m a.s.l. These are merely theoretical limits. In extreme cases, lake levels could drop under the lower limit if there is more water evaporating than precipitating or flowing in the lake.

The result and discussion sections are also updated, and the new text presented below, in the answer on Comment 24.

> **Reviewer 1 Comment 20**
>
> P6 L9: ". . .of the 7 tested the bias. . ." change to ". . .of the 7 tested bias. . ."

**Response**

We removed "the" from the sentence. The resulting sentence is as follows:

WB closure is adhered with two of the 7 tested bias correction methods described by Gudmundsson et al. (2012).

> **Reviewer 1 Comment 21**
>
> P6 L26: "Results of the evaluation WBM run are compared. . ." change to "Results of the evaluation WBM runs are compared. . ."

**Response**

We replaced "run" by "runs".

> **Reviewer 1 Comment 22**
>
> P7 L1: "Table 1. Table showing the a and b parameters of the linear. . ." change to "Parameters a and b of the linear. . ."

**Response**

We changed the table caption accordingly. It was not clear from the text that this table is only showing the parameters of the linear parametric transformation of the inflow term. We added also columns for the lake precipitation and evaporation term and moved the table to the appendix section:

An overview of the $a$ and $b$ parameters generated per WB term for the different simulations can be found in table B1 in the appendix.

This results in following new appendix section:

**Overview of the parameters used in the linear parametric transformation**

Table C1 shows the $a$ and $b$ calibration parameters for the different CORDEX simulations used in the linear parametric transformation to bias correct the lake precipitation, evaporation and inflow terms of the WB.

Table 1: Parameters $a$ and $b$ of the linear parametric transformation of the WB terms for the different CORDEX simulations (Eq. 3).

| RCM | Driving GCM | Lake precipitation | | Lake evaporation | | Inflow | |
| --- | --- | --- | --- | --- | --- | --- | --- |
| | | $a$ $(10^{-3})$ | $b$ | $a$ $(10^{-3})$ | $b$ | $a$ $(10^{6})$ | $b$ |
| CCLM4-8-17 | CNRM-CM5 | 0.65 | 0.350 | 2.30 | 0.476 | 27.0 | 0.257 |
| CCLM4-8-17 | EC-EARTH | 1.68 | 0.574 | 2.82 | 0.536 | 38.1 | 0.315 |
| CCLM4-8-17 | HadGEM2-ES | 1.69 | 0.425 | 2.88 | 0.339 | 47.6 | 0.404 |
| CCLM4-8-17 | MPI-ESM-LR | 1.20 | 0.342 | 2.05 | 0.556 | 36.6 | 0.261 |
| CRCM5 | MPI-ESM-LR | 1.06 | 0.810 | 1.71 | 0.676 | -16.5 | 0.705 |
| CRCM5 | CanESM2 | -1.54 | 0.850 | 2.86 | 0.655 | -15.0 | 0.647 |
| RACMO22T | EC-EARTH | -0.01 | 0.528 | 1.95 | 0.206 | -31.1 | 2.041 |
| RACMO22T | HadGEM2-ES | 0.43 | 1.823 | 2.98 | 0.944 | -16.0 | 2.225 |
| HIRHAM5 | EC-EARTH | 1.04 | 1.630 | 1.59 | 0.424 | 0.90 | 0.730 |
| RCA4 | CanESM2 | 1.78 | 0.926 | 1.78 | 0.658 | 16.0 | 0.759 |
| RCA4 | CM5A-MR | 1.81 | 0.860 | 1.86 | 0.629 | 24.4 | 0.691 |
| RCA4 | CNRM-CM5 | 1.95 | 0.982 | 1.78 | 0.643 | 27.6 | 0.916 |
| RCA4 | EC-EARTH | 1.64 | 0.794 | 1.66 | 0.706 | 19.0 | 0.657 |
| RCA4 | GFDL-ESM2M | 2.04 | 0.878 | 1.73 | 0.699 | 36.8 | 0.652 |
| RCA4 | HadGEM2-ES | 2.40 | 1.125 | 2.70 | 0.433 | 36.5 | 1.177 |
| RCA4 | MIROC5 | 1.73 | 0.885 | 2.00 | 0.593 | 23.9 | 0.767 |
| RCA4 | MPI-ESM-LR | 1.79 | 0.889 | 1.76 | 0.657 | 21.8 | 0.812 |
| RCA4 | NorESM1-M | 2.04 | 0.994 | 1.81 | 0.657 | 31.3 | 0.946 |
| RCA4 | CSIRO-Mk3-6-0 | 1.74 | 1.048 | 1.89 | 0.648 | 22.9 | 0.971 |
| REMO2009 | HadGEM2-ES | 0.16 | 0.425 | 2.90 | 0.241 | 33.5 | 0.525 |
| REMO2009 | MPI-ESM-LR | 1.23 | 0.639 | 2.50 | 0.519 | 42.2 | 0.814 |
| REMO2009 | EC-EARTH | 1.76 | 0.923 | 2.59 | 0.703 | 40.7 | 1.013 |
| REMO2009 | CM5A-LR | 0.73 | 0.414 | 2.81 | 0.271 | 19.7 | 0.453 |
| REMO2009 | GFDL-ESM2G | 0.51 | 0.415 | 2.58 | 0.327 | 13.3 | 0.470 |
| REMO2009 | MIROC5 | 0.57 | 0.481 | 2.88 | 0.279 | 23.1 | 0.524 |
* * *
**Reviewer 1 Comment 23**

P8 L2: ". . .lake levels (figure 4a)." change to ". . .lake levels (Fig. 4a)."
* * *
**Response**

We updated the manuscript accordingly.

> **Reviewer 1 Comment 24**
>
> P9 L9-...: the whole section (3.3), giving the results for water levels and outflow, needs to be revised. According to my understanding water levels above 1136.5 m asl would lead to increasingly high outflows (there is no structure to withhold water) and the water level cannot reach values as given, e.g., in Fig. 8 (=> reaching up to 1180 m asl). It is not clear what (if any) digital elevation model is used. A water level of 1180 m asl would mean (considering the flat shores/wetlands close to the lake) that an enormous area (lake surface area > 100.000 km2?) would be flooded.

**Response**

We revised figures 9-10, section 3.3 ('Future lake level and outflow projections') and parts of the discussion section, now taking the physical constraints of the lake level range and the two new management scenarios into account (see comments 2, 18 and 19). A rise in lake level to the newly constrained upper boundary (1146.9 m), could still imply important floodings of the flat shores and wetlands. Investigating these flooding scenarios is however out of the scope of this paper.

Changes in the results section:

[revised manuscript text omitted]

**Reviewer 1 Comment 25**

P9 L26: ". . .relation between outflow and lake level" would recommend changing to ". . .relation between lake level and outflow" as lake level is the guiding dimension

**Response**

We adjusted this in the text.

**Reviewer 1 Comment 26**

P10 L10-13: ". . .LVB (Fig 3 and Fig. 4)." change to ". . .LVB (Figs. 3 and 4)."

**Response**

We updated the manuscript accordingly.

**Reviewer 1 Comment 27**

P10 L21-22: "The other CORDEX models have no lake model embedded." Why this information is only given in the discussion? It would help to understand better the deviations of the other models when giving already in "2. Data and methodology".

**Response**

We understand the issue raised by the reviewer. We agree that it will help the reader if the information about in which RCMs lakes are represented by a lake module and added this in the methodology section:

> In CORDEX-Africa, there are currently simulations with six RCMs available (*CCLM4-8-17, CRCM5, HIRHAM5, RACMO22T, RCA4* and *REMO2009*) for the historical period (1950-2005) and the three RCPs (2006-2100). In the *CRCM5* and *RCA4* model, lakes are represented by a one dimensional lake model FLake (Samuelsson et al., 2013; Hernández-Díaz et al., 2012; Martynov et al., 2012), while the other RCMs have no lake model embedded.

**Reviewer 1 Comment 28**

P10 L33: ". . .Souverijns et al. (2016) (up to 0.5 mm-1 over the lake)." change to ". . .Souverijns et al. (2016) (up to 0.5 mm day-1) over the lake."

**Response**

We applied the suggestion in the text.

**Reviewer 1 Comment 29**

P12 L20: "Providing a constant hydropower supply, which implies a constant outflow,. . ." - The same outflow with a difference in the head of 2.5 m will lead to differences in the generated hydropower. . .

**Response**

We agree. See our response to comments 2, 18, 19 and 24.

> **Reviewer 1 Comment 30**
>
> P12 L22ff: Please check where you are referring to (or citing numbers for) electricity generation/demand and energy demand (energy also includes fuel for cars etc., the energy demand of a region/country therefore is higher than the electricity demand).

**Response**

We crossed-checked and clarified the distinction between energy and electricity in the text. This is indeed of great importance, a Uganda's electricity is mainly provided by hydropower, but this hydropower only contributes for 1% to Uganda's energy supply (Adeyemi and Asere, 2014).

> In Uganda, hydropower provides up to 90% of the electricity generated (Adeyemi and Asere, 2014). There is a rapidly growing gap between electricity supply and a rising demand, as the electricity consumption per capita in Uganda is among the lowest in the world. The Kiira and Nalubaale hydropower stations, managing Lake Victoria's outflow, are the largest power generators in the country (Adeyemi and Asere, 2014).

> **Reviewer 1 Comment 31**
>
> P13 L6 and 15: ". . .model uncertainty. . ." - Here you are referring to CLIMATE model uncertainty?

**Response**

Yes, we refer to climate model uncertainty. To clarify this we added the word climate in front of uncertainties on line 6 and 15.

> **Reviewer 1 Comment 32**
>
> P13 L25: ". . .a water balance model constructed developed for Lake Victoria. . ." change to ". . .a water balance model developed for Lake Victoria. . ."

**Response**

We removed the "constructed" from the sentence.

> ### Reviewer 1 Comment 33
>
> Appendix A P23-25: "...HIRHAM5 driven by EC-EARTH and CRCM5 driven by CanESM2 are not used due to the fact that they exhibit discrepancies between their historical and future simulations..." â please check if during the download, naming of files or at some other point (even check if an error occurred at the CORDEX homepage) an error occurred: According to Fig. A1 for the historical run (HIRHAM5) the precipitation is between 3-5 mm/day, while for scenarios (first years) according to Fig. A2 CRCM5 gives comparable values. According to Fig. A2 for the historical run (CRCM5) the precipitation is between 10-14 mm/day, while for scenarios (first years) according to Fig. A1 HIRHAM gives comparable values. Can it be that historical runs or scenarios between these two models were substituted (I see the same for evaporation and to a lesser extend for inflow)?

**Response**

We very much thank the reviewer for his/her notice of difference of the simulations. After a throughouh check, we found the inconsistency where HIRHAM5 and CRCM5 were swapped. This results in closing the historical and future simulation of CRCM5 (illustration 4). This simulation is taken into the analysis and table A1 is updated. The simulation of HIRHAM5 driven by EC-EARTH still contains inconsistencies, which we confirmed after thoroughly double checking the original simulations. The future simulation following RCP 4.5, has a discrepancy compared to the historical HIRHAM5-EC-EARTH simulation. This gap is enlarged when the bias correction is applied. Therefore, we exclude this simulation from the analysis.

We updated Appendix 1 with this information:

> From all available simulations, the simulation of HIRHAM5 driven by EC-EARTH following RCP 4.5 is not used because it exhibits discrepancies between its historical and future simulation. These discrepancies are nonphysical and inhibit the application of a bias correction.

The illustration 5 is however not included in the appendix, following comment 5 of reviewer 2.

[Figure]

Illustration 4: WB terms following historical and RCP 4.5 of CRCM5-CanESM2 simulations (without bias correction).

[Figure]

Illustration 5: WB terms following historical, RCP 4.5 and 8.5 of HIRHAM5-EC-EARTH simulations (without bias correction).

> **Reviewer 1 Comment 34**
>
> Appendix B P26 L3: ". . .condition to compared the WBM simulations. . ." change to
> ". . . condition to compare the WBM simulations. . ."

**Response**

We applied the suggestion.

> **Reviewer 1 Comment 35**
>
> Appendix B P26 L6: "To account for the missing days, 5 extra days are added for every
> 72 days in the year, starting after the 36th day." â please describe exactly how this is
> done;guess that "1 extra day is added each 72 day, starting at the 36th day." as shown
> in the table below day of year (orig.) 36 108 180 252 324 360 day of year (corr.) 37 110
> 183 256 329 365

**Response**

We updated the description of correcting the simulation for the 5 missing days:

> To account for the missing days, 5 extra days are added for each 72 days in the
> year, starting after the 36th day. The index of these 5 extra days within each year
> are given in table 2.

Table 2: Indices where extra days are added per year

| Original index     | 36 | 108 | 180 | 252 | 324 |
|--------------------|----|-----|-----|-----|-----|
| Index of added day | 37 | 210 | 183 | 256 | 329 |

> **Reviewer 1 Comment 36**
>
> Appendix C P26-28 L18: ". . .compare figs. C1, ..." change to ". . .compare Figs. C1,
> ...", also in the titles of Figs. C1, C2, C3 and C4 "As in Fig. . . ."

**Response**

We replaced the small by capital letters.

> **Reviewer 1 Comment 37**
>
> P28 L6: ". . .through ESGF.Wim Thiery was..." change to ". . .through ESGF. Wim
> Thiery was..."

**Response**

We adjusted the manuscript.

> ### Reviewer 1 Comment 38
>
> Please check the References, e.g.: P30, L2: "Adeyemi, K. O. and Asere, A. A.: a Review..." change to "Adeyemi, K. O. and Asere, A. A.: A Review..." P31, L20-21: "Mayaux, P., Massart, M., Cutsem, C. V., Cabral, a., Nonguierma, a., Diallo, O., Pretorius, C., Thompson, M., Cherlet, M., Defourny, P., Vas- concelos, M., Gregorio, a. D., Grandi, G. D., and Belward, a.:..." change to " Mayaux, P., Massart, M., Cutsem, C. V., Cabral, A., Nonguierma, A., Diallo, O., Pretorius, C., Thompson, M., Cherlet, M., Defourny, P., Vasconcelos, M., Gregorio, A. D., Grandi, G. D., and Belward, A.:..." P32 L23-24: is this a book chapter (editor, publisher?) or a journal article? P33, L1-2: "Vanderkelen, I., Lipzig, N. P. M. V., and Thiery, W.: Modelling the water balance of Lake Victoria (East Africa), part 1 : observational" change to "Vanderkelen, I., Lipzig, N. P. M. V., and Thiery, W.: Modelling the water balance of Lake Victoria (East Africa), part 1: observational" and include full reference (page no. etc.)

**Response**

We updated the reference list. The reference to Vanderkelen et al., 2018a, will be adjusted by the Copernicus office, if the two papers would be published together.

**2   Reviewer 2**

> **Reviewer 2 Comment 1**
>
> The paper presents an interesting study, very relevant from the point of view of future management of the Lake Victoria system. Messages of relevance are formulated in the paper. However, it is relatively difficult to discover these messages, as the paper unnecessarily focuses on the process of arriving at them rather than on the messages themselves. The focus on the process is not really justified, as no new approaches are used, nor methodologies are developed. The value of the paper is in the messages, description of methodology is necessary only to the extent that makes the messages defensible.
>
> There is an important consequence to the above. It is well known that GCMs and RCMs have biases. The purpose of the paper is not to evaluate performance of a set of RCMs in the Victoria Lake Basin, but to assess impacts of future climate on Lake Victoria's water balance. A large section of the paper, however, deals with RCM biases and evaluation of RCM performance in LVB. In my opinion, that section should be presented only if the results of evaluation were used to select a subset of models/simulations to be used in further analyses.

**Response**

We thank reviewer 2 for this critical view on the manuscript. Following the comment, we aimed to shift the focus from the methodologies to the messages of the paper by stressing the key messages:

In the abstract:

[revised manuscript text omitted]

Furthermore, we shortened the methodology section describing the bias correction methods. We removed the first paragraph describing bias correction in general and moved the explanation about the quantile-quantile bias correction method to the appendix. Also the paragraph in the results section of the original manuscript, which discusses the effect of the two bias correction methods is moved to the appendix section. In this way, the main part of the manuscript does not contain an elaborated description of the different bias correction methods, but an interested reader can find them in the appendix:

> Next to bias correction method using a linear parametric transformation (see section 2.4), WB closure was adhered with a second method which is the non-parametric quantile mapping method, a common approach for statistical transformation (e.g. Panofsky and Brier, 1968; Wood et al., 2004; Boé et al., 2007; Themeßl et al., 2011; Themeßl et al., 2012). Following Gudmundsson et al. (2012) and Boé et al. (2007), this method uses the Cumulative Density Function (CDF) based on the empirical quantiles from the observed variable to transform the modelled variable. First, the cumulative density functions of the three WB terms following each historical simulation in the overlapping period (the reference simulations) are matched with the cumulative density function of the WB terms from the observational WBM (observations). This generates a correction function, relating the quantiles of both distributions. Next, this correction function is used to unbias the WB term simulations for the whole simulation period quantile by quantile (Boé et al., 2007).

Furthermore, to make the distinction between the discussion of model quality and water management, we divided the discussion section in two subsections: 'Model quality and 'Water management and climate change'. We shortened the first section about the model evaluation from 1.5 pages to less than 1 page. The paragraph in the original discussion section dealing with the comparison of the two bias corrected methods is moved to the appendix section:

> In this study, applying a bias correction on the WB terms of the CORDEX simulations was necessary to be able to make lake level and outflow projections, as subsetting as not possible. RCMs are often bias corrected, as their simulations inhibit errors (Christensen et al., 2008; Teutschbein and Seibert, 2013; Maraun et al., 2010; Themeßl et al., 2012; Lange, 2018). Both linear parametric transformation and the quantile mapping bias correction methods are used. The advantage of the first is the simplicity and transparency of the method (Teutschbein and Seibert, 2013). The quantile mapping method on the other hand, is a non-parametric method and is able to correct for errors in variability as well (Themeßl et al., 2011). Yet, no substantial differences could be noted between the resulting projections of both methods, which supports that there is no single optimal way to correct for RCM biases (Themeßl et al., 2011). It is however important to consider the limitations concerning the bias correction methods. In both methods, each WB term is corrected independently, whereas biases may not be independent among the terms, which may be important in the context of climate change (Boé et al., 2007). The consistency between the variables could be preserved by using a more sophisticated method using a multivariate bias correction (Cannon,

2017; Vrac and Friederichs, 2015). However, Maraun et al. (2017) showed that bias correction could lead to improbable climate change signals and cannot overcome large model errors.

We also shortened and moved the paragraph about the comparison of the lake precipitation signal with the study of Souverijns et al. (2016) from the discussion to the results section:

The model simulations with a moderate increase or decrease also vary in sign. The ensemble mean projects an small increase of precipitation, which is not in line with the decreasing lake precipitation signal reported by Souverijns et al. (2016).

> **Reviewer 2 Comment 2**
>
> Bias correction based on 12 years of data is not appropriate, unless the authors are able to show that in the studied region decadal and multidecadal variability is very low.

**Response**

We thank the reviewer for raising this issue. Originally, the bias correction is calculated for the period 1993 to 2005, based on the overlapping period between the observational WBM of Vanderkelen et al. (2018) (1993-2014) and the historical CORDEX simulations (1951-2005). The analysis period in the observational WBM is constraint by the availability of the DAHITI lake level data (from 1992 on), while evaporation is calculated based on an average year taken from CCLM[2] and both lake precipitation and inflow are calculated with PERSIANN-CDR precipitation data. This PERSIANN-CDR dataset is however available from 1983 onwards. Therefore, it is possible to extend the bias correction period with 10 years, now ranging from 1983 to 2005 (22 years). The bias correction period could not be prolonged after 2005, because it is restricted by the period of the historical CORDEX simulations, on which the bias correction is calculated. Therefore, we recalculated the bias corrected WB terms based on a 22-year bias correction period, ranging from 1983-2005 and updated the bias correction methodology section accordingly:

This is done for the overlapping period of 22 years ranging from 1983 until 2005.

We updated all figures and numbers with the updated bias correction using the new calibration period. These changes have no influence on the messages of the paper.

> **Reviewer 2 Comment 3**
>
> The reason to consider the four outflow scenarios should be described better. They are not compatible with each other in that they represent totally different management approaches rather than a quantitative modification of a particular approach. It would make sense to present them if the authors were targeting a question of most efficient or effective or robust management approach under changing climate. But they are not. The justification of using the four is actually very weak and conceptually incorrect. The fact that in the past Agreed Curve was not adhered to does not mean that the other three are preferred alternatives.

**Response**

The outflow management scenarios were defined to illustrate the effect on lake levels using totally different approaches, as we started from the research question what the influence is of different management scenarios on future lake levels of Lake Victoria under a changing climate. However, we agree that the rationale behind the choice of the scenarios was lacking. Therefore we rephrased the first paragraph of the methodology section on dam management, stressing the policy objectives behind the scenarios. Two scenarios are redesigned for the electricity policy objective (aiming a constant hydropower production).

> Considering the known deviations of water release from the Agreed Curve during the period 2000-2006 (Kull, 2006; Vanderkelen et al., 2018), future outflow is subject to uncertainty. Therefore, we start from three main policy objectives concerning the environmental conservation, navigation reliability and constant electricity generation to determine future dam management scenarios. These policy objectives lead to four idealized dam management scenarios: (i) managing outflow following the Agreed Curve, reflecting natural conditions by mimicking natural outflow, (ii) managing outflow so that the lake level remains constant, to keep the lake accessible for fishing boats from the harbors in shallow bays, and (iii) managing outflow to provide a constant supply of hydropower from the dams: one scenario prescribing the historical mean production of hydropower and the other an elevated hydropower production, reflecting the supply needed to meet the rising power demand in Uganda (Adeyemi and Asere, 2014). These scenarios are highly simplified and reflect very different management; they were chosen to investigate the effect of extreme dam management scenarios on the lake level fluctuations of Lake Victoria. Each scenario will thereby be applied with lake levels constrained to their physical boundaries.

We also updated this in the introduction:

> Last, the future evolution of the water level of Lake Victoria under various climate change scenarios is investigated, together with the role of different human management strategies at the outflow dams based on three policy objectives (environment, navigation and electricity).

> Reviewer 2 Comment 4
>
> PERSIANN rainfall, and COSMO-CLM are hardly what one would understand as "observations". Aren't there "real" observational data available? Rain gauges? Met stations? if not - then why these products were chosen? why not CHIRPS? or TRMM/GPM?

**Response**

The observational WB model and a elaborate description of the products used can be found in the first paper of this two paper series (Vanderkelen et al., 2018). In previous WB studies, lake precipitation is often assessed based on lake shore stations, but these are not capable of fully capturing precipitation on the lake, as precipitation on the lake is almost doubled compared to the surrounding land by the presence of the lake itself (Thiery et al., 2015). Therefore, we used PERSIANN-CDR, a state-of-the-art gridded precipitation product based on remote sensing data adjusted to the Global Precipitation Climatology Project (GPCP) monthly product version 2.2. The latter product is derived from rain gauge data.

There are no measurements of evaporation over Lake Victoria. Previous studies therefore used simple empirical models to calculate evaporation which require multiple climatic input data and which are very sensible to this input data (Yin and Nicholson, 1998). Another option would be to use global evaporation datasets like LandFlux-EVAL and the GLobal Land Evaporation Amsterdam Model (GLEAM), but these products do not resolve evaporation from large water bodies like Lake Victoria (Mueller et al., 2013; Martens et al., 2017). Therefore we used the output of the COSMO-CLM$^2$ regional climate model. Although this is indeed an output from a reanalysis downscaling, we believe that this is the most accurate available data on Lake Victoria evaporation, and preferable over other models computing evaporation. For more details on this matter, we refer to Vanderkelen et al. (2018).

To check the validity of our choice to use PERSIANN-CDR as a reference precipitation dataset, we conducted the same analysis with the TRMM-3B42 precipitation product for the period 1998-2008. Illustration 6 shows that the annual precipitaiton over the Lake Victoria Basin has a very similar pattern for PERSIANN-CDR and TRMM. The amount of precipitation in the northeastern part of the basin is however much higher for the PERSIANN-CDR product, consistent with Overshooting Top satellite imagery (Thiery et al., 2016, 2017). We used the WBM to calculate the resulting lake levels when using the TRMM-3B42 precipitation product (both for the lake precipitation term and in the calculation of the inflow term). Using the TRMM-3B42 product however leads to a WB which is not in equilibrium and lake levels which decrease (illustration 7). This can roughly be attributed to the decreased inflow, following the lower precipitation amount in the northeastern part of the basin showed in illustration 6. This analysis strengthens our choice to use the PERSIANN-CDR product for precipitation over the Lake Victoria Basin.

[Figure]

Illustration 6: Annual precipitation over Lake Victoria (inner polygon) and its basin (outer polygon) for the products PERSIANN-CDR (1993-2014) and TRMM - 3B42 (1998-2008).

[Figure]

Illustration 7: Observed and modelled lake level with TRMM-3B42 used as precipitation observation product (similar to fig. 9 of Vanderkelen et al. (2018)).

**Reviewer 2 Comment 5**

there is no need to show the HIRHAM discrepancy in the appendix. Itâs enough if it is stated clearly in the text.

**Response**

The HIRHAM and CRCM5 discrepancies are clarified in the response on comment 33 of reviewer 1. We therefore omitted figure showing the discrepancy in the manuscript. Consequently, the revised Appendix section *A: Details on used CORDEX simulations* does not contain any figure.

> ### Reviewer 2 Comment 6
>
> what is the outer polygon in Figure 1? NB. a study area map showing the lake, main rivers, catchment and location of dams would be beneficial.

**Response** We added a description of the polygons to figures 1 and 2:

We see the benefit of a study area map, and included the same map included in the first part of this two part paper series (illustration 8):

> The outflow of the lake is controlled by the Nalubaale and Kiira dams for hydropower generation, located in Jinja (fig. 1).

[Figure]

Illustration 8: Map of Lake Victoria and its basin with surface heights from the Shuttle Radar Topography Mission (SRTM). Taken from Vanderkelen et al. (2018).

From this figure, it should be clear to the reader that the inner polygon in figures 1 and 2 represents Lake Victoria Basin and the outer polygon represents the basin.

> ### Reviewer 2 Comment 7
>
> fig 6: in the caption, climate change signal is described as a difference between historical and future. In such a case, is change with positive sign an increase into the future? or a decrease into the future?

**Response**

We thank the reviewer for raising this point. In the caption of figures 6 and 7, a confusing sentence was used, while it was well explained in the manuscript itself: *"This is achieved by computing the difference between the future (mean of the period 2071-2100) and the historical (mean of the period 1971-2000) simulations (Fig. 6)"*. It means that if the climate change signal is positive, there is an increase in the future and vice versa. We made this consistent in the caption of all figures (see also illustrations 9 and 10).

**Reviewer 2 Comment 8**

fig 6 and fig 7 show essentially the same information. Only fig 7 could be used. Bias correction could change signal, but only under very specific conditions. Also, showing 95% confidence interval would help assessing the strength of signal.

Changes in precipitation and evaporation should be expressed as a percent (relative change), not as a difference (absolute change). Change expressed as difference in a situation of strong biases (offsets) may result in obtaining negative values when that change is compared to observations.

**Response**

We adjusted figures 6 and 7 to show the relative climate change signal and added the 75% confidence intervals (illustrations 9 and 10).

We think however that figure 6 and 7 are sufficiently different to be both included in the manuscript as they show the overall decrease in magnitude of the relative climate signal, together with signal changes for precipitation in RCP 4.5 and 8.5 and evaporation in RCP 8.5. Therefore both figures are kept in the manuscript. (But I am not completely convinced to keep both) Following the updated figures, we updated the paragraphs in the manuscript describing these figures:

> For some simulations, lake precipitation demonstrates a strong decrease (e.g. CCLM4-8-17 driven by EC-EARTH, CRCM5 driven by CanESM2 and REMO2009 driven by EC-EARTH) while other simulations show a strong increase (e.g. HIRHAM5 driven by EC-EARTH and REMO2009 driven by CM5A-LR). The model simulations with a moderate increase or decrease also vary in sign. ... For the three WB terms, the width of the 95% confidence intervals is larger for strong climate change signals.

[Figure]

Illustration 9: Barplots showing the relative projected climate chance following RCP 2.6, 4.5 and 8.5 for lake precipitation (a-c), lake evaporation (d-f) and inflow (g-i) for the CORDEX simulations without bias correction. The climate change signal is defined as the difference between future (2071-2100) and the historical (1971-2000) simulations. The whiskers indicate the 95% confidence interval of the change based on the 30-year annual difference.

[Figure]

Illustration 10: Barplots showing the relative projected climate chance following RCP 2.6, 4.5 and 8.5 for lake precipitation (a-c), lake evaporation (d-f) and inflow (g-i) for the CORDEX simulations bias corrected with the linear parametric transformation. The climate change signal is defined as the mean difference between the future (2071-2100) and the historical (1971-2000) simulations. The whiskers indicate the 95% confidence interval of the change based on the 30-year annual difference.

> **Reviewer 2 Comment 9**
>
> Table 1 is not necessary, and values of coefficient "a" shown in it seem unrealistic. If model has shown no bias, value of a would be 0. Table 1 has values in the order of 10000000.

**Response**

We agree with the reviewer and moved table 1 to the appendix section. Moreover, as stated in the response on Comment 22 of Reviewer 1, the original table only showed the $a$ and $b$ parameters for the inflow term. Therefore, we expanded the table so that it also shows the $a$ and $b$ parameters for the lake precipitation and evaporation terms.

The values of parameter $a$ in the original table are perceived 'unrealistic' because they reflect changes in the inflow term, which has an order of magnitude of $10^7$ to $10^8$ $m^3$ day$^{-1}$. This explains the large values for parameter $a$. The same is true for the lake precipitation and evaporation terms with an order of magnitude of $10^{-3}$ m day$^{-1}$. In the updated appendix table we accounted for these different orders of magnitude (see also Reviewer 1, Comment 22).

> **Reviewer 2 Comment 10**
>
> what is Pm in eq. 3?

**Response**

We clarified $P_m$ in the manuscript as follows:

[revised manuscript text omitted]

Martens, B., Miralles, D. G., Lievens, H., Van Der Schalie, R., De Jeu, R. A., Fernández-Prieto, D., Beck, H. E., Dorigo, W. A., and Verhoest, N. E. (2017). GLEAM v3: Satellite-based land evaporation and root-zone soil moisture. *Geoscientific Model Development*, 10(5):1903–1925.

Martynov, A., Laprise, R., Sushama, L., Winger, K., Šeparović, L., and Dugas, B. (2012). Reanalysis-driven climate simulation over CORDEX North America domain using the Canadian Regional Climate Model, version 5: Model performance evaluation. *Climate Dynamics*, 41(11-12):2973–3005.

Mueller, B., Hirschi, M., Jimenez, C., Ciais, P., Dirmeyer, P. A., Dolman, A. J., Fisher, J. B., Jung, M., Ludwig, F., Maignan, F., Miralles, D. G., McCabe, M. F., Reichstein, M., Sheffield, J., Wang, K., Wood, E. F., Zhang, Y., and Seneviratne, S. I. (2013). Benchmark products for land evapotranspiration: LandFlux-EVAL multi-data set synthesis. *Hydrology and Earth System Sciences*, 17(10):3707–3720.

Nicholson, S. E. (2016). An analysis of recent rainfall conditions in eastern Africa. *International Journal of Climatology*, 36(1):526–532.

Nicholson, S. E. (2017). Climate and climatic variability of rainfall over eastern Africa. *Reviews of Geophysics*, 55(3):590–635.

Nikulin, G., Jones, C., Giorgi, F., Asrar, G., Büchner, M., Cerezo-Mota, R., Christensen, O. B., Déqué, M., Fernandez, J., Hänsler, A., van Meijgaard, E., Samuelsson, P., Sylla, M. B., and Sushama, L. (2012). Precipitation climatology in an ensemble of CORDEX-Africa regional climate simulations. *Journal of Climate*, 25(18):6057–6078.

Otieno, V. O. and Anyah, R. O. (2013). CMIP5 simulated climate conditions of the Greater Horn of Africa (GHA). Part II: Projected climate. *Climate Dynamics*, 41(7-8):2099–2113.

Panofsky, H. and Brier, G. W. (1968). *Some Apllications of Statistics to Meteorology*. The Pennsylvania State University Press, Philadelphia.

Rowell, D. P., Booth, B. B. B., and Nicholson, S. E. (2015). Reconciling Past and Future Rainfall Trends over East Africa. *Journal of Climate*, 28:9768–9788.

Samuelsson, P., Gollvik, S., Jansson, C., Kupiainen, M., Kourzeneva, E., and Berg, W. J. V. D. (2013). The surface processes of the Rossby Centre regional atmospheric climate model (RCA4). (157).

Schwatke, C., Dettmering, D., Bosch, W., and Seitz, F. (2015). DAHITI - An innovative approach for estimating water level time series over inland waters using multi-mission satellite altimetry. *Hydrology and Earth System Sciences*, 19(10):4345–4364.

Sene, K. J. (2000). Theoretical estimates for the influence of Lake Victoria on flows in the upper White Nile. *Hydrological Sciences Journal*, 45(August):125–145.

Souverijns, N., Thiery, W., Demuzere, M., and van Lipzig, N. P. M. (2016). Drivers of future changes in East African precipitation Drivers of future changes in East African precipitation.

Sutcliffe, J. and Parks, Y. (1999). The Hydrology of the Nile. *IAHS Special Publication*, 5(5):192.

Sutcliffe, J. V. and Petersen, G. (2007). Lake Victoria: derivation of a corrected natural water level series. *Hydrological Sciences Journal*, 52(September 2015):1316–1321.

Teutschbein, C. and Seibert, J. (2013). Is bias correction of regional climate model (RCM) simulations possible for non-stationary conditions. *Hydrology and Earth System Sciences*, 17(12):5061–5077.

Themeßl, M. J., Gobiet, A., and Heinrich, G. (2012). Empirical-statistical downscaling and error correction of regional climate models and its impact on the climate change signal. *Climatic Change*, 112(2):449–468.

Themeßl, M. J., Gobiet, A., and Leuprecht, A. (2011). Empirical-statistical downscaling and error correction of daily precipitation from regional climate models. *International Journal of Climatology*, 31(10):1530–1544.

Thiery, W., Davin, E. L., Panitz, H.-J., Demuzere, M., Lhermitte, S., and van Lipzig, N. (2015). The Impact of the African Great Lakes on the Regional Climate. *Journal of Climate*, 28(10):4061–4085.

Thiery, W., Davin, E. L., Seneviratne, S. I., Bedka, K., Lhermitte, S., and Van Lipzig, N. P. (2016). Hazardous thunderstorm intensification over Lake Victoria. *Nature Communications*, 7.

Thiery, W., Gudmundsson, L., Bedka, K., Semazzi, F. H., Lhermitte, S., Willems, P., van Lipzig, N. P. M., and Seneviratne, S. I. (2017). Early warnings of hazardous thunderstorms over Lake Victoria. *Environmental Research Letters*, 12(7):2–5.

Vanderkelen, I., Lipzig, N. P. M. V., and Thiery, W. (2018). Modelling the water balance of Lake Victoria (East Africa), part 1 : observational analysis. *Hydrol. Earth Syst. Sci*, discussion(January).

Vrac, M. and Friederichs, P. (2015). Multivariate-intervariable, spatial, and temporal-bias correction. *Journal of Climate*, 28(1):218–237.

Williams, K., Chamberlain, J., Buontempo, C., and Bain, C. (2015). Regional climate model performance in the Lake Victoria basin. *Climate Dynamics*, 44(5-6):1699–1713.

Wood, A. W., Leung, L. R., Sridhar, V., and Lettenmaier, D. P. (2004). Hydrologic implications of dynamical and statistical approaches to downscaling climate model outputs. *Climate Change*, 62:189–216.

Yin, X. and Nicholson, S. (1998). The water balance of Lake Victoria. *Hydrological Sciences Journal*, 43(2):789–811.